# Two-Stage Learning of Stabilizing Neural Controllers via Zubov Sampling and Iterative Domain Expansion

**Haoyu Li**\*   **Xiangru Zhong**\*   **Bin Hu**    **Huan Zhang**
University of Illinois Urbana-Champaign
haoyuli5@illinois.com, xiangru4@illinois.edu
binhu7@illinois.com, huan@huan-zhang.com

## Abstract

Learning-based neural network (NN) control policies have shown impressive empirical performance. However, obtaining stability guarantees and estimates of the region of attraction of these learned neural controllers is challenging due to the lack of stable and scalable training and verification algorithms. Although previous works in this area have achieved great success, much conservatism remains in their frameworks. In this work, we propose a novel two-stage training framework to jointly synthesize a controller and a Lyapunov function for continuous-time systems. By leveraging a Zubov-inspired region of attraction characterization to directly estimate stability boundaries, we propose a novel training-data sampling strategy and a domain-updating mechanism that significantly reduces the conservatism in training. Moreover, unlike existing works on continuous-time systems that rely on an SMT solver to formally verify the Lyapunov condition, we extend state-of-the-art neural network verifier $\alpha,\beta$-CROWN with the capability of performing automatic bound propagation through the Jacobian of dynamical systems and a novel verification scheme that avoids expensive bisection. To demonstrate the effectiveness of our approach, we conduct numerical experiments by synthesizing and verifying controllers on several challenging nonlinear systems across multiple dimensions. We show that our training can yield region of attractions with volume $5 - 1.5 \cdot 10^5$ times larger compared to the baselines, and our verification on continuous systems can be up to $40 - 10{,}000$ times faster compared to the traditional SMT solver dReal. Our code is available at `https://github.com/Verified-Intelligence/Two-Stage_Neural_Controller_Training`.

## 1   Introduction

Learning-based neural network control policies have demonstrated great potential in complex systems due to their high expressiveness [19, 53]. However, the applications of these neural controllers in safety-critical real-world physical systems raise concerns since they typically lack stability guarantees. A promising method for quantifying safety is estimating the region of attraction (ROA), which refers to the set of initial states from which a system is guaranteed to converge to a desired equilibrium point. The computation of the ROA is often formulated as the search for an appropriate Lyapunov function satisfying some algebraic conditions over its sublevel sets [26, 41, 31]. Due to the highly complex and nonlinear nature of neural networks, searching for a Lyapunov function is not easy. Many existing works that are able to make stability guarantees about a system have some restrictions on the problem structure, for example, requiring linear or polynomial parameterizations of the system, where sum-of-squares technique [37, 47, 30, 7] can be applied. To enable certification for general nonlinear systems, many recent works explore data-driven methods to synthesize a neural certificate [8], such as

---

\*Equal contribution.

39th Conference on Neural Information Processing Systems (NeurIPS 2025).

Lyapunov functions [4, 6, 55, 44, 48, 35, 24, 34, 29, 23], barrier Functions [3, 32, 7, 10, 25, 9, 16, 17], and contraction metrics [38, 40, 39, 36, 22], to certify the system's stability.

In this work, we aim to synthesize neural controllers of continuous-time systems together with a Lyapunov certificate. Unlike typical machine learning tasks, this problem requires that the Lyapunov and boundary conditions hold over the *entire continuous domain of interest*, rather than statistically achieving high accuracy on a discrete test set. However, since at each iteration only a small number of points can be selected, the problem forces us to adopt specialized strategies both to pick the training domain and to sample training data from the domain. We argue that these two are the key factors for the conservativeness issue observed in many previous works. For selecting the training domain, many previous works simply select a fixed domain of interest and enforce the Lyapunov condition over it. While some previous works have proposed a new formulation to overcome the conservativeness by introducing an extra term to verify forward invariance and allow the ROA to touch the training domain nontrivially [48, 35], we argue that this does not fully solve the conservativeness issue and leaves the training domain selection a hard hyperparameter to tune.

In addition to the training domain, training data selection also plays an important role. Besides simple random sampling that typically does not suffice for certification, previous works dealing with similar problems [4, 6, 55, 48] typically achieve co-learning with a counterexample-guided (CEGIS) approach, which generates counterexamples of the Lyapunov condition as the training data, either using a verifier [4, 6] such as an SMT or MILP solver or a gradient-based approach [44, 48]. While these approaches have led to great success, such approaches can typically only refine upon already worked controller/Lyapunov function pair and thus rely on carefully tuned LQR-/RL-based initializations [4, 44, 48]. This limitation makes these works both conservative in their estimation of the ROA and also very difficult to apply to new systems without much domain-specific knowledge.

In our work, we propose a two-stage training framework with carefully designed curriculum domain expansion and training sample selection that can yield significantly less conservative ROA estimation inspired by Zubov theorem [56, 24, 18, 28], which provides a characterization of the true ROA with a PDE. During the first stage, we propose a Zubov-inspired training data sampling strategy by combining points from both inside and outside of the current ROA estimation to better guide the learning. Moreover, we eliminate the conservativeness introduced by the training domain by dynamically expanding it using trajectory information. This stage can provide us an almost-working controller and Lyapunov function that we can refine upon. The second stage then involves counterexample-guided learning (CEGIS) to eliminate all the counterexamples inside the ROA estimation and thus obtain a verifiable Lyapunov certificate. Importantly, our training also fully utilizes the physics-informed Zubov loss and can eliminate the conservatism introduced by the inner approximation nature of the normal Lyapunov function. As we shall demonstrate in our numerical experiments, our approach can robustly produce controllers with significantly larger ROAs compared to previous state-of-the-art approaches in all the benchmarks with various control input limits.

As we deal with continuous-time systems, verification over Jacobian vector products also presents a challenge. Previous works addressing the Lyapunov stability verification for continuous-time systems typically rely on SMT-, MIP-, or SDP-based solvers [4, 1, 24, 5, 49, 14, 5] such as dReal [15] or Z3 [11]. Inspired by the success in [48, 35], we also utilize the state-of-the-art neural network verification tool $\alpha,\beta$-CROWN [52, 33, 45, 46, 43, 51] for formally verifying Lyapunov stability. The key challenge is to handle the Jacobian-related operators in dynamical systems, which have not been discussed or implemented in existing neural network verifiers. To ensure verification tightness, we design new linear relaxations for these Jacobian operators in continuous time system learning such as Tanh and Sigmoid. Furthermore, we introduce a novel verification scheme to dynamically adjust the specifications when noticing counterexamples without incurring much computational cost to avoid expensive bisections as in [48, 24].

To summarize, our main contributions include:

• We introduce a novel two-stage Zubov-based training framework with carefully designed curriculum training domain expansion and training sample selection that significantly reduces the conservatism during training.

• We customize and strengthen the advanced neural network verification tool $\alpha,\beta$-CROWN with the ability of handling many commonly used Jacobian-related operators in continuous-time systems. Moreover, we propose a novel verification algorithm that can effectively avoid expensive bisections without sacrificing on soundness.

- We demonstrate through numerical experiments that our training can yield region of attraction with volume $5 - 1.5 \cdot 10^5$ times larger than the baselines. Moreover, across the systems under evaluation which we can formally verify, our verification achieves $40 - 10,000$ times speedup compared to the previous SMT solver dReal.

## 2 Background and Problem Statement

We consider a nonlinear continuous-time state-feedback control system

$$\dot{x} = g(x, \pi(x)) = f(x) \tag{1}$$

where $x$ is the system state, $g$ denotes the open-loop dynamics, $u$ denotes the controller, and $f$ denotes the closed-loop nonlinear dynamics. We denote the solution of the system starting with initial condition $x = x_0$ by $\phi(t, x_0)$, and denote the equilibrium state and the control input as $x^*/u^*$ respectively. We denote the sublevelset $\{x : f(x) \leqslant c\}$ of some function $f$ as $f^{\leqslant c}$ and correspondingly $f^{=c}$ for the levelset. Our objective is to jointly search for a neural network control policy $u_\theta$ together with a Lyapunov function $V_\theta$ that can certify the stability of the closed-loop dynamics (1) according to the Lyapunov theorem. The goal is to learn a controller that maximizes the region of attraction (ROA), which has the formal definition below.

**Definition 2.1** (Region of Attraction (ROA) [21]). Region of attraction for the system (1) is the set $\mathcal{R}$ such that $\lim_{t \to \infty} \phi(t, x_0) = x^*$ for all $x_0 \in \mathcal{R}$.

**Theorem 2.2** (Lyapunov Theorem [26, 21]). *Given a forward invariant set $\mathcal{S}$ containing the equilibrium, if there exists a function $V : \mathcal{S} \to \mathbb{R}$ such that we have*

$$\begin{aligned}
V(x^*) &= 0 \\
V(x) &> 0 \quad (\forall\, x \in \mathcal{S} \backslash \{x^*\}) \\
\nabla V(x)^\top f(x) &< 0 \quad (\forall\, x \in \mathcal{S})
\end{aligned} \tag{2}$$

*then $\mathcal{S}$ is a subset of the region of attraction.*

Therefore, the objective of this work can be formalized as the optimization problem

$$\max_{\pi_\theta, V_\theta} \mathrm{Vol}(\mathcal{S}), \quad \text{subject to Theorem (2.2)} \tag{3}$$

To maximize the possible ROA, we adopt the Zubov theorem, which states that a function that satisfies a stricter set of constraints than the Lyapunov condition can express the real underlying ROA $\mathcal{R}$.

**Theorem 2.3** (Zubov Theorem [56, 24, 18]). *Consider the nonlinear dynamical system in (1) and we assume that $x^* = 0$ without loss of generality. Let $\mathcal{A} \subset \mathbb{R}^n$ and assume that there exists a continuously differentiable function $W : \mathcal{A} \to \mathbb{R}$ and a positive definite function $\Psi : \mathbb{R}^n \to \mathbb{R}$ satisfying the following conditions,*

$$W(0) = 0 \text{ and } 0 < W(x) < 1, \; \forall x \in \mathcal{A} \backslash \{0\} \tag{4}$$

$$W(x) \to 1 \text{ as } x \to \partial \mathcal{A} \text{ or } \|x\| \to \infty \tag{5}$$

$$\nabla W(x)^\top f(x) = -\Psi(x)(1 - W(x)), \quad \forall x \in \mathcal{A} \tag{6}$$

*Then $x^* = 0$ is asymptotically stable with the region of attraction is exactly $\mathcal{A}$.*

Here, equation (6) is called the Zubov equation and can be utilized as the physics-informed loss during our training. It is noticed that there exists a ground truth value for the Zubov function that can be obtained through the following theorem.

**Theorem 2.4** (Liu et al. [24]). *We adopt the convention that divergence of integration yields the value $\infty$. Then for any $a, p > 0$, $W(x) = \tanh(a \int_0^\infty \|\phi(t, x)\|^p \, dt)$ solves the Zubov equation equation $\nabla W(x)^\top f(x) = -a(1 + W(x))(1 - W(x))\|x\|^p$.*

We refer the readers to [24] for more theoretical details. This theorem enables us to obtain the ground truth value of $W$ during the training without solely depending on the PDE residual loss, making the training process more robust, and can provide better learning dynamics for the Lyapunov function.

## 3 Methods

### 3.1 Learning of Neural Controller and Lyapunov Function

In this section, we introduce our training pipeline for synthesizing the neural controller and a corresponding Lyapunov function. As we have discussed, the selection of the training domain and training samples are the most crucial factors for successful training. We shall see that the Zubov theorem provides us not only the physics-inspired loss, but also a principled way of data selection. During training, we parameterize the controller and Lyapunov function as

$$u_\theta(x) = c \cdot \tanh(\mathrm{NN}_c(x) - \mathrm{NN}_c(x^*) + \tanh^{-1}(\frac{u^*}{c})), \quad V_\theta(x) = \mathrm{sigmoid}(\mathrm{NN}_v(x)) \quad (7)$$

This parametrization ensures that we have $0 \leqslant V_\theta \leqslant 1$ and the crucial property $u_\theta(x^*) = u^*$, which ensures that the origin is also an equilibrium for the closed-loop system. Moreover, this parametrization constrains the control input lie in the range $[-c, c]$. Sometimes, if a non-symmetric limit $[0, c]$ is more natural, we append another ReLU layer to $u_\theta$. This operation can also be applied to each coordinate separately to support mixed control-input limits.

#### 3.1.1 ROA Estimation Stage

The first stage of colearning which we call ROA estimation stage, aims to set up a mostly working controller and an estimate of the Lyapunov function that can be further refined. As we have discussed, in this stage the most important aspect is the selection of training data and training domain.

**Zubov Guided Sampling:** Intuitively, a good training data selection scheme should mix the data points inside and outside of the running region of attraction. Selecting only "outside" points risks collapsing the sublevel set back toward the origin, while sampling exclusively "inside" offers no signal to enlarge it. In low dimensions, a well-chosen training box plus random sampling can approximate this mix, but as dimensionality grows, the ROA occupies a vanishing fraction of the domain, and naive draws will almost always only select the points outside of the ROA. For example, in the cartpole system, the ROA we found at the end of

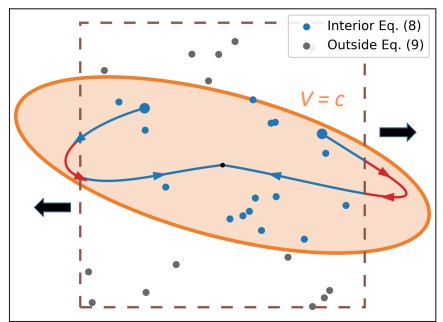

Figure 1: An illustration of the key idea used in the ROA estimation stage. Notice the balanced training sample selection from both in and outside of the ROA estimation $V^{\leqslant c}$. The training domain is also dynamically expanded to include all the states along a convergent trajectory.

training only occupies less than $1\%$ of the full training domain, and this will only be more pronounced in even higher-dimensional systems, and therefore naive sampling is infeasible.

In the traditional Lyapunov function framework, it is difficult to obtain this running estimation of the ROA. Zubov theorem, on the contrary, characterizes the exact region of attraction by the sublevel set $V^{\leqslant 1}$. Intuitively, in our training pipeline, since our parametrization already deems that $V$ cannot exactly equal 1, we propose picking the training data by selecting a batch of points lying in $V_\theta^{\leqslant c}$ and another batch with $V_\theta(x)$ as close to 1 as possible, with $c \approx 1$. To achieve this, we select data by performing projected gradient descent (PGD) [27] with the following two objectives

$$L_{\mathrm{interior}}(\theta) = \mathrm{ReLU}(V_\theta(x) - c) \quad (8)$$
$$L_{\mathrm{outside}}(\theta) = |V_\theta(x) - 1|. \quad (9)$$

The first interior loss will push points to the sublevel set $V_\theta^{\leqslant c}$ and the outside loss will push points further away from the region of attraction as shown in Figure 1. A detailed discussion on how to pick $c$ is presented in Appendix A.4.

**Dynamic Training Domain Expansion:** While previous works attempt to resolve the conservativeness introduced by a fixed training domain by allowing the ROA to touch non-trivially with it, we argue that the best way is instead to dynamically expand the domain during training. We thus

propose to utilize the trajectory information to dynamically update the training box. We start from a relatively small training region and periodically perform a trajectory simulation starting from the points inside the current ROA estimation $V_\theta^{\leqslant c}$ with the same $c$ in the data selection. We then expand the training region to include the farthest states that the converged trajectories can reach. To also benefit simple systems where all trajectories converge to zero in a monotone fashion, we further enlarge the box uniformly for the final training domain. If there is no convergent trajectory for many iterations, we manually enlarge the box for further exploration. This approach simultaneously resolves conservativeness in ROA posed by fixed domain and also resolves the challenging forward invariance issue. A visualization of our approach can be found in Figure 1, where the original blue training domain is expanded further to incorporate the states along the convergent trajectories. The domain updating algorithm can be found in Algorithm 2 in appendix.

**Loss Function:** The training loss can be easily adapted from the Zubov theorem. The objective of the training is to train the Lyapunov function and controller to encourage $V_\theta$ to behave more like a Zubov function. Therefore, naturally we could minimize $V_\theta(0)$ and train on the PDE residual of (6) and the data loss (2.4). In the handling of data loss, previous works such as [42] often simulates the trajectory till convergence to compute $\int_0^\infty \|\phi(t,x)\|^p \, dt$. This, however, would lead to very lengthy training and add difficulty in the implementation of efficient batched operations since different trajectories have different convergence speeds. We thus propose to replace this term using a Bellman-type dynamic programming approach. We note that the following rewrite is possible:

$$
\begin{aligned}
W(x) &= \tanh(a \int_0^\infty \|\phi(t,x)\|^p \, dt) \\
&= \tanh(a \int_0^T \|\phi(t,x)\|^p \, dt + \tanh^{-1}(W(\phi(T,x)))).
\end{aligned}
\tag{10}
$$

With this approach, only simulation to time $T$ is required. In practice, we found that this term can benefit training even if a very small $T$ like $T = 0.01s$ is adopted. We also utilized a similar strategy as in [42] to decouple the learning of the Lyapunov function and controller using the actor-critic framework and impose the boundary condition $V_\theta(y) \approx 1$ for $y \in \partial(2\Omega)$ to guide the early phase learning. Moreover, to ensure that the second CEGIS stage is numerically stable, we minimize $V_\theta(0)$ only with a hinge-type loss. Let $\epsilon > 0$ be a small positive number. The final loss function is a weighted combination of the following components:

$$
L_{\text{zero}}(\theta) = \text{ReLU}\left(V_\theta^2(0) - \epsilon\right),
\tag{11}
$$

$$
L_{\text{pde}}(\theta) = \frac{1}{k_1} \sum_{i=1}^{k_1} \left(\nabla V_\theta(x_i)^\top f(x_i, \text{detach}(u(x_i))) + a(1 + V_\theta(x_i))(1 - V_\theta(x_i))\|x_i\|^p\right)^2,
\tag{12}
$$

$$
L_{\text{data}}(\theta) = \frac{1}{k_2} \sum_{i=1}^{k_2} \left(V_\theta(x_i) - \tanh(a \int_0^T \|\phi(t,x_i)\|^p \, dt) - \tanh^{-1}(V_\theta(\phi(T,x_i)))\right)^2,
\tag{13}
$$

$$
L_{\text{controller}}(\theta) = \frac{1}{k_1} \sum_{i=1}^{k_1} \text{detach}(\nabla V_\theta(x_i))^\top f(x_i, u(x_i)),
\tag{14}
$$

$$
L_{\text{boundary}}(\theta) = \frac{1}{k_3} \sum_{i=1}^{k_3} \left(V_\theta(y_i) - 1\right)^2,
\tag{15}
$$

where $k_1, k_2, k_3 > 0$ represents the number of data points. Here, the PDE loss minimizes the PDE residual (6) by optimizing the Lyapunov function, whereas the controller loss optimizes the controller akin to a Sontag-style formula. Here $x_i$ are training data sampled by our novel sampling scheme, and $y_i$ are the boundary points from $\partial(2\Omega)$. More details about the loss function design can be found in appendix A.3.

### 3.1.2 CEGIS-based Refinement

Many previous works [4, 44, 48] start their training with a counterexample-guided approach (CEGIS [2]). However, without a good initialization, the counterexamples generated cannot be easily eliminated and will accumulate through the training and cause failure eventually, which is

shown in the ablation studies Table 2. We found that our first stage can yield a great initialization for CEGIS-based training. The controller after the pretraining stage is already almost working, meaning that almost all trajectories starting within $V_\theta^{\leqslant c}$ will successfully converge to the equilibrium. However, as we have discussed, no certification can be given at this time since random samples typically are not sufficient to guarantee the condition holds over the entire continuous domain. Therefore, another stage of CEGIS-based finetuning is performed to ensure the certifiability of the system. To find counterexamples, we follow the previous works as in [48, 44] to use a cheap gradient-based PGD attack with the objective

$$L_{\text{cex}}(x) = \min(\nabla V_\theta(x)f(x), V_\theta(x) - c', c - V_\theta(x)) \tag{16}$$

Maximizing this objective leads to potential $x_{\text{cex}}$ that violates the Lyapunov condition in the band $V_\theta^{\leqslant c}\backslash V_\theta^{\leqslant c'}$. Here we require a lower $c'$ since there is no guarantee that $V_\theta(0) = 0$, and so near the origin there must exist some small violations. However, we can choose this $c'$ to be just slightly larger than $V_\theta(0)$. These counterexamples then serve as the dataset for the CEGIS training. We then eliminate these counterexamples with the loss function

$$L_{\text{cegis}}(\theta; x_{\text{cex}}) = \text{ReLU}(\nabla V_\theta(x_{\text{cex}})f(x_{\text{cex}})) \tag{17}$$

where it enforces the Lyapunov condition. Moreover, the overall shaping of the Lyapunov function should mostly be formed at the ROA estimation stage due to the presence of Zubov-inspired loss and should thus be only slightly refined at the CEGIS stage. Therefore, we denote $v_0$ as the value of $V_\theta(0)$ prior to the CEGIS finetuning, and apply the regularization term

$$L_{\text{reg}}(\theta) = \max(\text{ReLU}(v_0 - V_\theta(0)), \text{ReLU}(V_\theta(0) - c')) + \sum_{x\in\partial\Omega} |V_\theta(x) - 1| \tag{18}$$

with the aim of preventing the levelset of $V_\theta$ from changing drastically during CEGIS. The first term thus stabilizes the levelset structure near the origin by restricting $V_\theta(0)$ to stay close to the original value, whereas the second term stabilizes the levelset near the boundary of the training domain and encourages the ROA to stay away from this boundary to prevent trajectories from escaping from it.

---

**Algorithm 1** Training of Neural Controller and Lyapunov Function

---

**Require:** Network Parameters $\theta$, Starting Domain $\Omega$, Maximum Iterations $M_1, M_2$, Integration Time $T$ and Discretizations Timesteps $dt$, Learning rate $\alpha$, Number of Training Points $N$, Domain Update Frequency $\gamma$, Number of Trajectories $N_T$, Lyapunov Upper Threshold $c$, Domain Expansion Factor $\beta$. PGD steps $P$ and step size $\alpha_1$, Finetuning learning rate $\alpha_2$ and epochs $K$
1: **for** $i = 1$ **to** $M$ **do**                                        ▷ Stage 1: ROA Estimation Stage
2:      **if** $\gamma \mid i$ **then**
3:          $\Omega \leftarrow$ UPDATEDOMAIN$(\theta, \Omega, T, dt, N_T, c, \beta)$                       ▷ See algorithm 2
4:      $x \leftarrow N$ data points from (8) and (9)
5:      $x_{\text{boundary}} \leftarrow N$ points from $\partial(2\Omega)$
6:      $\theta \leftarrow \theta - \alpha\nabla L(\theta, x, x_{\text{boundary}}, T, dt)$                           ▷ See loss function section.

7: Dataset $\leftarrow \varnothing$                                           ▷ Stage 2: CEGIS Refinement
8: **for** $i = 1$ **to** $M$ **do**
9:      $x \leftarrow P$ points from $\Omega$
10:      **for** $j = 1$ **to** $P$ **do**
11:          $x \leftarrow x + \alpha_1 \nabla L_{\text{cex}}(x)$                               ▷ See equation (16)
12:      Dataset $\leftarrow \{$Dataset$, x\}$             ▷ Buffer has length limit. See appendix .
13:      **for** $j = 1$ **to** $K$ **do**
14:          $\theta \leftarrow \theta - \alpha_2 \nabla(\beta_1 L_{\text{cegis}}(\theta; \text{Dataset}) + \beta_2 L_{\text{reg}}(\theta))$          ▷ See section 3.1.2
15:      **if** No CEX in dataset for at least $n$ steps **then**
16:          break
17: **return** Trained parameters $\theta$, Final Training Domain $\Omega$

---

## 3.2 Verification

**Criterion:** Since there is no guarantee that $V_\theta(0) = 0$, counterexamples near the origin are unavoidable. As the Lyapunov condition enforces the Lyapunov function value to decrease along trajectories,

we propose to exclude an unverifiable region defined as a sublevel set of the Lyapunov function. If the excluded set is small enough, this verification suffices for most practical needs. Formally, we propose the verification criterion

**Theorem 3.1.** *Let $0 < c_1 < c_2$. Suppose we have*

$$x \in (V^{\leq c_2} \backslash V^{\leq c_1}) \cap \Omega \longrightarrow \nabla V(x) \cdot g\big(x, \pi(x)\big) < 0, \tag{19}$$

$$x \in \partial\Omega \cap V^{\leq c_2} \longrightarrow g\big(x, \pi(x)\big) \cdot \vec{n}(x) < 0, \tag{20}$$

*where $\vec{n}(x)$ is the outer normal vector of $\partial\Omega$ at $x$. Then both $V^{\leq c_1}$ and $V^{\leq c_2}$ are* forward invariant, *and every trajectory initialized in $V^{\leq c_2}$ enters the smaller set $V^{\leq c_1}$ in finite time.*

The proof of this theorem can be found in appendix section B.1. Here equation (19) states the normal Lyapunov condition has to hold in the band $\{x : c_1 \leq V(x) \leq c_2\}$ whereas equation (20) states that the trajectory cannot escape through the boundary. Compared to previous works that simply exclude a rectangle near the origin [48], our verification scheme is more principled since it also provides some guarantees inside the unverifiable region. In systems where formal certification over the full domain is challenging, verifying the condition within a thin band where $c_2 - c_1 < \epsilon$ for arbitrarily small $\epsilon$ still allows one to certify the forward invariance of the sublevel set $V^{\leq c_1}$.

**Formal Verification:** For formal verification, instead of the commonly used SMT solver dReal in previous works [15, 4, 24], we hope to use a state-of-the-art neural network verification framework, such as $\alpha,\beta$-CROWN [52, 33, 45, 46, 43, 51, 54], to rigorously verify the Lyapunov condition through the tool's efficient linear bound propagation and branch-and-bound procedures. However, several **key challenges remain** for the application of $\alpha,\beta$-CROWN. First, since the continuous-time Lyapunov condition involves Jacobian-vector products, it is essential to support efficient and tight bound propagation for the Jacobian of various dynamical system operators, which were previously not supported in any NN verifiers. Second, even after applying a CEGIS procedure, small counterexamples can still arise near the levelset boundaries at $c_1$ and $c_2$. To identify the tightest verifiable values of $c_1$ and $c_2$, a lengthy and expensive bisection is typically required, as commonly done in prior work [48, 24].

In our work, we extend $\alpha,\beta$-CROWN with the capability to efficiently and tightly bound the Jacobian of many commonly used operators such as Tanh, Sigmoid. We implemented new linear relaxations that are much tighter compared to naively treating their Jacobian as composite functions of already supported operators, where the details can be found in the appendix section B.3. Furthermore, we enable our verifier to dynamically update $c_1, c_2$ during verification to avoid expensive bisection while remain sound. Intuitively, during the branch-and-bound process, we pick out those domains currently with the worst bound and try to find counterexamples in those domains. If counterexamples are found, we update $c_1$ and $c_2$ to exclude the counterexamples. Notice that after this update, all the verified subproblems during branch and bound still remain sound and do not need to be verified again. This effectively saves the time needed for the repetitive verification needed if $c_1$ and $c_2$ are to be found by bisection. The full verification algorithm can be found in appendix section B.4.

**Evaluation Schemes:** Compared to discrete-time systems, where the final verification is performed on the discrete Lyapunov condition $\Delta V(x) = V(g(x, u(x)) - V(x) < 0$ through domain. Continuous-time systems require formally verifying the Jacobian vector product (19). The large number of multiplications is challenging for current verifiers since it is hard to bound multiplication well with a linear relaxation. Therefore, even with the strongest verification tool, verification is hard to scale to higher dimensions due to bound tightness issues. We thus define three evaluation schemes we considered. As system dimensionality increases, we must rely increasingly more on empirical checks.

1. **Formal Verification:** This scheme implies the condition in theorem (3.1) is formally verified throughout the domain with a verifier.

2. **PGD Evaluation:** In this scheme, we use a strong PGD attack to attempt finding counterexamples of the condition in theorem (3.1). If counterexamples are not found, we say the empirical PGD evaluation succeeds.

3. **Trajectory Evaluation:** In this scheme, we randomly sample trajectories from the sublevel set $V^{\leq c_2}$ and determine whether the trajectories converge to the desired equilibrium.

We can notice that these schemes are more and more empirical but are also more and more scalable to higher dimensional systems. The formal verification scheme, even though it is the strongest verification and can imply the other two schemes, can be hard to scale up. The empirical PGD

evaluation can scale up better, but still fails on our hardest 12-dimensional system. However, the trajectory based evaluation can be easily scaled up to any dimension. While this sampling approach provides no formal guarantees, several previous works utilize this technique to validate the learned controller [50, 10, 47].

## 4 Experiments

**Setups:** We demonstrate the effectiveness of our methods in several challenging benchmarks including four 2D systems (Van-Der-Pol, Double-Integrator, Inverted Pendulum, Path Tracking), one 4D system (Cartpole), three 6D systems (PVTOL, 2D Quadrotor, Ducted Fan), and a 12D system (3D Quadrotor). System details follow prior works [44, 48, 35] and are also provided in appendix C.2. We compare our training framework against baselines from DITL [44], Yang et al. [48], and Shi et al. [35]. We note that all of them work with discrete-time systems. We focus on comparing to these baselines due to limitations in the existing continuous-time methods. The works [4, 42, 13] incorrectly handle the condition $u(x^*) = u^*$, so that the origin fails to be the equilibrium for most systems. Further discussions and examples are included in the appendix C.1. However, to ensure fair comparison, we still run and present the comparison with Fossil 2.0 [13] for those systems that their method handle correctly. For each method, we run the training/verification pipeline five times with different seeds. We compare training success rates and ROA volumes, where the latter is estimated following previous works [44, 35] as the fraction of points within the ROA sublevel set among a dense sample from the full domain. All models undergo the three evaluation steps from section 3.2 in reverse order. More experimental details can be found in appendix C.

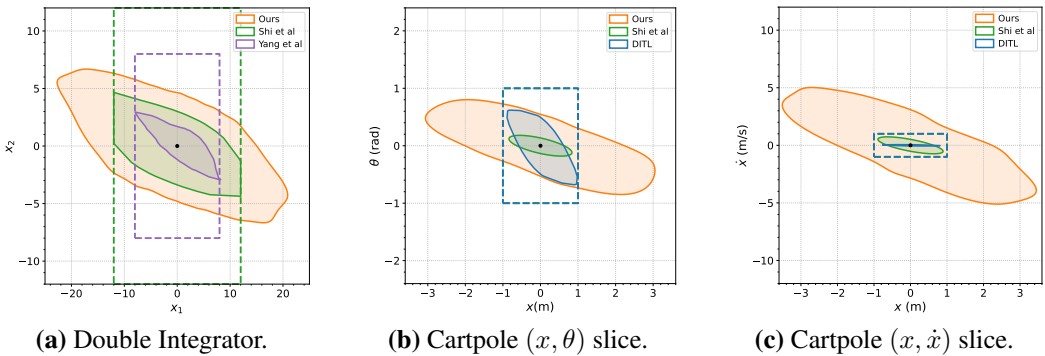

**(a)** Double Integrator.          **(b)** Cartpole $(x, \theta)$ slice.          **(c)** Cartpole $(x, \dot{x})$ slice.

Figure 2: Region-of-Attraction (ROA) for Double Integrator and Cartpole compared to baselines. The dotted box shows the baseline's training box being used. For the double integrator system, it is clear that the initial training box is limiting the baseline's potential but is not constraining us.

**Two and Four Dimensional Systems:** On all the 2D systems, our methods robustly produce much larger ROAs compared to existing baselines. On both the path tracking and the inverted pendulum system, our training framework yields **5** times bigger ROA compared to the baselines as shown in table 1. Moreover, on the more challenging systems such as inverted pendulum with small torque, our method is the only one that can get perfect success rate. Visualizations of the learned ROA for Double Integrator can be seen in the left of Figure 2. We also test our method on a more challenging Cartpole system with four dimensions. Compared to the baselines, we obtain an ROA with more than **330×** larger than the previous state of the art, as can be seen in Table 1. The visualizations for different projections of the ROA for Cartpole can be found on the right of Figure 2. The rest visualizations and details can be found in the Appendix C.6.

**Six Dimensional Systems:** We also implement our method on three higher dimensional systems: PVTOL, 2D Quadrotor, and Ducted Fan. For each system, our training pipeline yields a much larger ROA robustly. Compared to the baselines, we obtain an ROA with more than **150000×** larger than the previous state of the art on average for 2D quadrotor and more than **380×** larger for the PVTOL system as can be seen in Table 1. In the ducted fan case, we are the only method that can achieve an empirically verifiable ROA. We choose one projection to visualize for each of the system given in Figure 4. More visualizations can be found in the Appendix C.6. Due to the difficulty of continuous-time system verification, formal verification for these three systems is challenging.

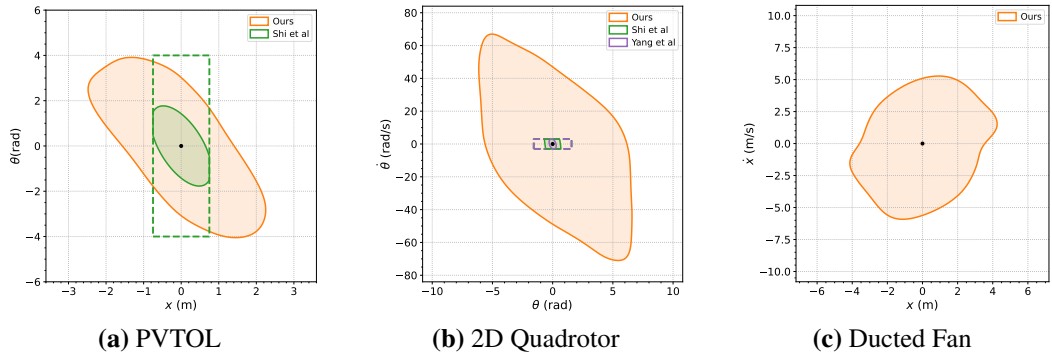

**(a)** PVTOL          **(b)** 2D Quadrotor          **(c)** Ducted Fan

Figure 3: Slice of the ROA for the 6D benchmark systems compared to baselines.

Table 1: Success rates (Succ) and ROA volume (ROA) across methods. Dashes (–) denote systems that are not already covered by the baseline and we were unable to obtain successful runs. The "Scheme" column indicates the evaluation scheme for each system as described in section 3.2: Formal = formal verification; PGD = empirical PGD evaluation; Trajectory = trajectory evaluation. "B" in a system implies the Big-torque setting, and "S" implies Small-torque.

| System | Eval Scheme | Fossil [13] | | DITL [44] | | Yang et al. [48] | | Shi et al. [35] | | Ours | |
|---|---|---|---|---|---|---|---|---|---|---|---|
| | | ROA | Succ | ROA | Succ | ROA | Succ | ROA | Succ | ROA | Succ |
| Van-der-Pol | Formal | $3.05 \pm 0.58$ | 60% | – | – | $1.36 \pm 1.57$ | 80% | $20.2 \pm 18.3$ | 80% | $\mathbf{57.6 \pm 3.4}$ | 100% |
| Double Int | Formal | $258.1 \pm 36.3$ | 80% | – | – | $18.18 \pm 16.19$ | 60% | $130.3 \pm 3.9$ | 60% | $\mathbf{302.5 \pm 10.7}$ | 100% |
| Pendulum B | Formal | $106.5 \pm 23.5$ | 80% | $61 \pm 31$ | 100% | $70.6 \pm 12.2$ | 100% | $487.5 \pm 58.5$ | 80% | $\mathbf{2946.5 \pm 149.1}$ | 100% |
| Pendulum S | Formal | $285.2 \pm 53.3$ | 80% | – | – | $217.34 \pm 6.07$ | 60% | $306.3 \pm 48.7$ | 40% | $\mathbf{1169.2 \pm 124.5}$ | 100% |
| Path Tracking B | Formal | – | – | $9 \pm 3.5$ | 100% | $24.06 \pm 0.29$ | 100% | $15.3 \pm 8.9$ | 60% | $\mathbf{122.0 \pm 3.7}$ | 100% |
| Path Tracking S | Formal | – | – | – | – | $14.86 \pm 0.18$ | 100% | $12.5 \pm 6.5$ | 100% | $\mathbf{73.8 \pm 12.5}$ | 100% |
| Cartpole | Formal | – | – | $0.021 \pm 0.012$ | 100% | – | – | $0.9266$ | 20% | $\mathbf{306.1 \pm 54.2}$ | 100% |
| PVTOL | PGD | – | – | – | – | – | – | $49.87 \pm 3.91$ | 80% | $\mathbf{(1.91 \pm 0.23) \cdot 10^4}$ | 100% |
| 2D Quadrotor | PGD | – | – | – | – | $2.33 \pm 0.47$ | 100% | $44.53 \pm 18.38$ | 80% | $\mathbf{(6.64 \pm 4.67) \cdot 10^6}$ | 100% |
| Ducted Fan | PGD | – | – | – | – | – | – | – | – | $\mathbf{(4.31 \pm 1.86) \cdot 10^4}$ | 100% |
| 3D Quadrotor | Trajectory | – | – | – | – | – | – | – | – | $\mathbf{(1.17 \pm 0.64) \cdot 10^9}$ | 100% |

However, empirical verification is confirmed for these systems, and in 2D Quadrotor, we can formally verify the forward invariance of the sublevelset $V^{\leqslant 0.2}$ using Theorem 3.1.

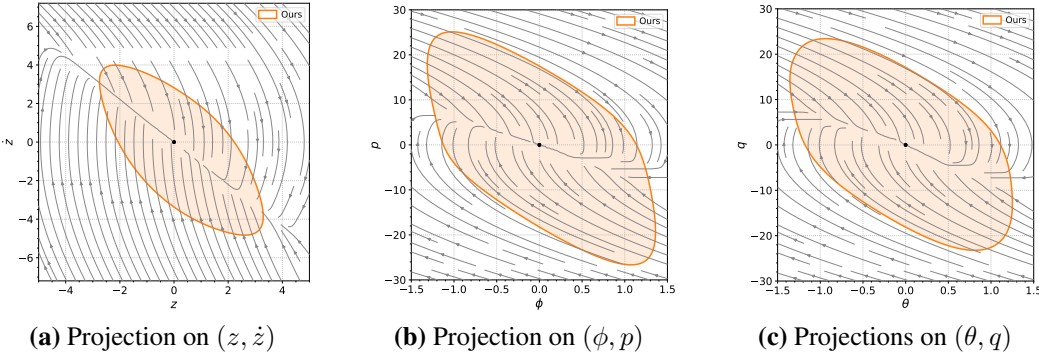

**(a)** Projection on $(z, \dot{z})$      **(b)** Projection on $(\phi, p)$      **(c)** Projections on $(\theta, q)$

Figure 4: ROA estimation for 3D quadrotor. As we can see from the flow map, all the trajectories in our ROA estimation converges to the origin.

**Twelve Dimensional System:** Finally, we test our training pipeline in a 12 dimension 3D quadrotor system, in which none of the previous methods successfully give any ROA estimation. In this high dimensional case, we found that our pretraining framework works smoothly and can consistently yield a working controller. However, we found that CEGIS in higher dimensions does not work perfectly and we oftentimes found that counterexamples can still be located using longer PGD with more restarts even if a prolonged CEGIS is already performed. However, the empirical success of our approach is confirmed by sampling over $10^6$ trajectories inside the levelset $V^{\leqslant c_2}$ and confirm that they all converge. The flow map in Figure 4 also confirms this.

Table 2: Success rates of different training schemes under three verification regimes.

| System | Training scheme | Evaluation scheme (% success) | | |
|---|---|---|---|---|
| | | Formal | Empirical-PGD | Trajectory |
| Cartpole | Two-stage with Random sampling | 0 | 0 | 0 |
| | ROA-Estimation stage only (Zubov + Domain Expansion) | 0 | 0 | 100% |
| | CEGIS only | 0 | 0 | 0 |
| | Ours (full pipeline) | 100% | 100% | 100% |
| 2-D Quadrotor | Two-stage with Random sampling | 0 | 0 | 0 |
| | ROA-Estimation stage only (Zubov + Domain Expansion) | 0 | 0 | 100% |
| | CEGIS only | 0 | 0 | 0 |
| | Ours (full pipeline) | – | 100% | 100% |

**Verification Time:** As we have discussed, our verifier struggles to verify higher-dimensional systems due to the bound tightness issue over Jacobian vector product. However, we shall note that our approach is already much faster compared to the previous state-of-the-art verifier dReal [15] used by many previous work [4, 24] for continuous time systems. In the 2D systems verification, our method yields $40 - 10000\times$ faster verification compared to dReal. And for the cartpole system, the dReal fails to terminate. The full verification time comparison can be found in Table 3.

**Ablation Studies:** We also investigate the effectiveness of our training design choices. We mainly focus on the effectiveness of our training sample selection scheme and the two-stage training by considering three training settings:

1. Two Stage training with domain expansion but with *random sampling*.
2. *Only the first ROA estimation stage* with Zubov inspired sampling and domain expansion.
3. *Only CEGIS* and domain expansion, no ROA estimation stage.

We then investigate the success rate of these three training schemes under different evaluations with two higher dimensional systems Cartpole and 2D quadrotor. The overall result can be found in Table 2. We can found that without our careful design choices, neither training Scheme 1 or Scheme 3 can yield any verifiable result. However, with our designed ROA estimation stage, even without CEGIS our trained controller and Lyapunov function can robustly pass the trajectory-based verification. Then together with CEGIS stage, our full training pipeline consistently yields controller/Lyapunov function that can be either formally verified or pass the PGD attack evaluation.

Table 3: Example verification time for each solver across our benchmark systems with formal verification. For 2D quadrotor, it is marked with a star due to only forward invariance of the set $V^{\leqslant 0.2}$ is verified by involving theorem 3.1 with $c_1 = 0.2, c_2 = 0.21$.

| Method | Van der Pol | Double Integrator | Pendulum B | Pendulum S | Path Tracking B | Path Tracking S | Cartpole | 2DQuad* |
|---|---|---|---|---|---|---|---|---|
| dReal | 39265.21 s | 359.27 s | 1479.55 s | 1709.19 s | 113.72 s | 142.20 s | – | – |
| $\alpha,\beta$-CROWN | 3.94 s | 3.00 s | 3.64 s | 3.94 s | 3.77 s | 3.67 s | 76443 s | 104614 s |

## 5 Conclusion

In this work, we propose a two stage training pipeline with novel training domain and training data selection for obtaining stabilizing controller with large region of attraction. Through extensive numerical experiments across various dimensions, we demonstrate that our training pipeline can achieve ROA volumes that are $5 - 1.5 \cdot 10^5$ bigger than the baselines. We also extend the state-of-the-art verifier $\alpha,\beta$-CROWN with the capability of performing bound propagation through many Jacobian operators and introduce novel verification schemes to avoid expensive bisection. We demonstrated that our verification can yield a $40 - 10,000$ times speedup compared to the previously most commonly used verifier dReal. Although our method achieves promising empirical performance regarding the ROA volume and dimensionality compared to existing baselines, formal verification is still challenging for higher dimensional systems. As future works, it will be interesting to explore more verifier-friendly training strategies and to further tailor and improve the verifier itself for this specific problem. Moreover, it would be interesting to extend the framework to more general control system settings, such as discrete-time or hybrid systems, slowly time-varying systems, or to synthesize and verify robust controller for systems with perturbations.

## Acknowledgment

H. Zhang is supported in part by the AI2050 program at Schmidt Sciences (AI2050 Early Career Fellowship) and NSF (IIS-2331967). B. Hu is generously supported by the AFOSR award FA9550-23-1-0732.

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

# A  Training Details

## A.1  Domain Update Algorithm

The full domain update algorithm can be found in algorithm 2. It first samples a batch of points from the sublevelset $V^{\leq c}$ by PGD with the objective (8), and then performs trajectory simulation starting from these points. The algorithm then pick out those convergent trajectories and update the training domain to include the farthest state possible these convergent trajectories can reach.

---

**Algorithm 2** UpdateDomain

---

**Require:** Controller $u_\theta$, Lyapunov Function $V_\theta$, Domain $\Omega$, Integration Time $T$ and Discretizations Timesteps $dt$, Number of Trajectories $N_T$, Lyapunov Upper Threshold $c$, Domain Expansion Factor $\beta$

1: $x_{\text{init}} \leftarrow N_T$ samples from the sublevel set $0 < V_\theta < c$                    ▷ Data from (8)
2: $\mathcal{C} \leftarrow$ Set of indices $j$ where trajectory $\phi(t, x_{\text{init}}^j)$ converges
3: **for** each dimension $d$ **do**
4:      $d_{\min} \leftarrow \min_{j \in \mathcal{C}, t \in [0,T]} [\phi(t, x_{\text{init}}^j)]_d$
5:      $d_{\max} \leftarrow \max_{j \in \mathcal{C}, t \in [0,T]} [\phi(t, x_{\text{init}}^j)]_d$
6:      $\Omega_d \leftarrow [d_{\min}, d_{\max}]$
7: $\Omega \leftarrow \beta \times \Omega$                                                          ▷ To handle monotonic trajectories.
8: **return** $\Omega$

---

## A.2  CEGIS Dataset

The dataset from line 12 of algorithm 1 will not grow without limitations. In practice, we fix a maximal size $k$ allowed for this dataset. After each iteration, we sort the dataset in descending order according to the violation of Lyapunov condition, and only keep the top $k$ samples.

## A.3  Loss Function

As we have discussed in the main text, the training loss is simply an adaptation of the Zubov theorem. The goal is to guide $V_\theta$ to behave more like the Zubov function through training. We adopt a similar loss function as in [42] and modify the data loss using the novel derivation (10). Moreover, to ensure that the second CEGIS stage is numerically stable, we minimize $V_\theta(0)$ only with a hinge-type loss. In the loss functions, $x_i$ are training data from our novel data selection (8) and (9). $y_i$ are boundary points from $\partial(2\Omega)$ and this boundary loss would help during the initial phase of learning by ensuring that the controller does not drive the trajectory to escape from at least two times the training domain. To ensure a balanced weight of the loss term and ease the hyperparameter tuning, we adopt loss weights that are learnable using the techniques presented in [20].

## A.4  Hyperparameters

Compared to previous works, our training framework significantly reduces the difficulty of hyperparameter tuning when adapting to new systems. For example, the training domain now does not require too much tuning, and the training almost always works when starting from a small domain. However, several key hyperparameters still require careful adjustment.

Firstly, the weights for each loss term must be chosen with care. In practice, we have found that the boundary loss and the zero loss benefit from higher weights compared to the other three loss terms. Conversely, assigning a relatively small weight to the PDE residual loss tends to yield better results. We hypothesize that this is because an overly dominant PDE loss introduces a complex optimization landscape, which can impede the learning of other critical components. Secondly, the integration involved in the data loss must be approximated using a numerical scheme. Both the discretization timestep and the number of simulation steps play an important role in ensuring stable and efficient learning. We found that a simple forward Euler scheme, with a small timestep such as $dt = 0.001\,\text{s}$ and approximately 50 simulation steps, is often sufficient. In this context, as long as $dt$ is not excessively small, a smaller $dt$ generally performs better than a larger one, since

the dynamics can exhibit stiffness when the controller is only partially learned. Moreover, during trajectory simulations used for domain updates, a smaller $dt$ and longer steps are recommended. Using a large timestep in this setting can cause otherwise convergent trajectories to be recognized as non-convergent. Finally, in the training data selection, the choice of $c$ is crucial. We typically fix it to be very close to 1, like $c = 0.95$. However, as the dimension of the system goes up, the Zubov equation might not be fitted well, so we reduce $c$ to be around 0.9 for better training stability.

## B    Verification Details

### B.1    Proof of theorem 3.1

*Proof.* Notice that if the trajectory is in $(V^{\leqslant c_2}\backslash V^{\leqslant c_1}) \cap \Omega$, the Lyapunov function value will keep decreasing along the trajectory. Indeed, by the mean value theorem, we have

$$V(x(t)) - V(x(0)) = t\dot{V}(x(\tau)) < 0 \tag{21}$$

First, we shall show that $V^{\leqslant c_2} \cap \Omega$ is forward invariant. Assume by contradiction that there exists $x \in V^{\leqslant c_2} \cap \Omega$ such that $\phi(T, x) \notin V^{\leqslant c_2} \cap \Omega$ for some $T > 0$. We define the first exit time as $T^* = \inf\{t : \phi(t, x) \notin V^{\leqslant c_2} \cap \Omega\}$. By continuity, we have $\phi(T^*, x) \in \partial(S_2 \cap \Omega)$, which satisfies

$$\phi(T^*, x) \in \partial(V^{\leqslant c_2} \cap \Omega) \subset (\partial\Omega \cap V^{\leqslant c_2}) \cup (\Omega \cap V^{=c_2})$$

If $\phi(T^*, x) \in (\partial\Omega \cap V^{\leqslant c_2})$, condition (20) says that it cannot exit through the boundary of $\Omega$. Otherwise, suppose $\phi(T^*, x) \in \Omega \cap V^{=c_2}$. We know by definition and continuity that on a small time interval $(T^* - \delta, T^*)$ we have $V(x(t)) < c_2$, which contradicts the decrease on Lyapunov function. Next, we show that on finite time the trajectory will enter $\Omega \cap V^{\leqslant c_1}$. Indeed, suppose for the contrary that the trajectory stays within $(V^{\leqslant c_2}\backslash V^{\leqslant c_1}) \cap \Omega$, the Lyapunov value will decrease indefinitely as we know by the above derivation $V(x(t_1)) < V(x(t_2))$ for $t_1 > t_2$, which cannot be possible by the definition of the set. Finally, we show that $S_1$ is forward invariant. With the same proof, it is easy to show that for any $0 < \epsilon < c_2 - c_1$ the set $V^{\leqslant c_1 + \epsilon}$ is forward invariant. Now if $S_1$ is not forward invariant, we know by definition that there exists $x \in V^{\leqslant c_1}$ and $t > 0$ such that $V(\phi(x, t)) = c_1 + \epsilon'$. This violates the forward invariance of $V^{\leqslant c_1 + \epsilon}$ for any $\epsilon < \epsilon'$, leading to a contradiction. $\square$

### B.2    The $\alpha,\beta$-CROWN verifier

$\alpha,\beta$-CROWN has been developed to be the state-of-the-art neural network verifier, where formally, the toolbox aims to certify

$$F(x) \leqslant 0 \text{ for all } x \in \mathcal{B} \tag{22}$$

with $F$ being a general computation graph and $B$ being a hyper-rectangle.

**High Level Intuition:** $\alpha,\beta$-CROWN couples fast *symbolic linear bound propagation* with *input-space branch-and-bound*. For a box $\mathcal{B}$, it constructs an affine upper bound of $F$ and provide certification based on the maximum value of this affine bound. If inconclusive, it *splits* $\mathcal{B}$ into sub-boxes and repeats. As boxes shrink, local linear relaxations tighten and nonlinearities stabilize, so on small sub-boxes the affine bound is often already exact. The full process is very efficient, as each bound computation essentially costs a backward pass; thus many sub-boxes can be processed in parallel (tens of thousands), enabling rapid pruning of large regions and certification with only coarse bounds on tiny boxes. Moreover, thanks to the tensorized nature of bound propagation and the flexibility of the branch-and-bound procedure, $\alpha,\beta$-CROWN can naturally handle general specifications with "or" and "and". In principle, $\alpha,\beta$-CROWN can handle all cases a traditional SMT solver, such as dReal, can handle, and in a much more scalable fashion. We refer the readers to the following papers for more details [52, 45, 45, 46, 43, 51].

For our application, since the condition in theorem 3.1 can be effectively rewritten as

$$(\nabla V(x) \cdot g(x, u(x)) < 0) \vee (V(x) > c_2) \vee (V(x) < c_2) \tag{23}$$
$$(g(x, u(x)) \cdot \vec{n}(x) < 0) \vee (x \notin \partial\mathcal{B}) \vee (V(x) > c_2), \tag{24}$$

and each term is a computation graph, $\alpha,\beta$-CROWN can efficiently verify this condition.

## B.3 Linear Relaxation of $\frac{d}{dx}\tanh(x)$ and $\frac{d}{dx}\text{sigmoid}(x)$

To compute the Jacobian of a neural Lyapunov function involving $\tanh(x)$ and $\text{sigmoid}(x)$, by the chain rule, we need to propagate the gradient through the operators $\frac{d}{dx}\tanh$ and $\frac{d}{dx}\text{sigmoid}$. Naturally, to use CROWN [52] to compute bounds for the Jacobian of the network, we need to linearly relax these operators so that we can propagate linear bounds through them. Given

$$\frac{d}{dx}\tanh(x) = 1 - \tanh^2(x),$$

$$\frac{d}{dx}\text{sigmoid}(x) = \text{sigmoid}(x)(1 - \text{sigmoid}(x)),$$

$$(25)$$

Finding linear relaxations of $\frac{d}{dx}\tanh$ and $\frac{d}{dx}\text{sigmoid}$ can be reduced to computing the composition of the linear relaxations of $\tanh(x)$, $x^2$, $\text{sigmoid}(x)$, and $x \cdot y$. The linear relaxations of these operators are already supported by auto_LiRPA [45]. However, in practice, this results in overly conservative relaxations. Thus, in our work, we directly derive a novel linear relaxation for these two gradient operators.

$\frac{d}{dx}\tanh(x)$ and $\frac{d}{dx}\text{sigmoid}(x)$ are "bell-shaped" functions having similar monotonicity and convexity/concavity properties, as shown in figure 5. Thus, we apply similar strategies finding linear relaxations for both of them. Suppose the function to be bounded is denoted as $\sigma(x)$, with the bounds of $x$ as $x \in [l, u]$. We aim to find a lower bound $\alpha^L x + \beta^L$ and an upper bound $\alpha^U x + \beta^U$, such that

$$\alpha^L x + \beta^L \leqslant \sigma(x) \leqslant \alpha^U x + \beta^U, \quad \forall x \in [l, u], \tag{26}$$

where parameters $\alpha^L, \beta^L, \alpha^U, \beta^U$ are dependent on $l$ and $u$. In our following discussion, for a "bell-shaped" functions $\sigma(x)$, we denote $\pm z_\sigma$ as the two inflection points (i.e. the solution to $\sigma''(x) = 0$), and we assume that $l + u \geqslant 0$ without loss of generality. In many cases, the linear bound is chosen to be a tangent line of $\sigma$, and thus is solely determined by the tangent point $d^L$ or $d^U$.

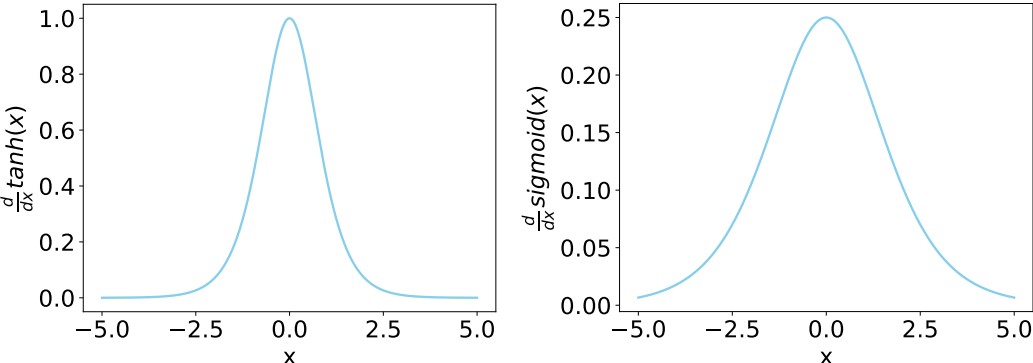

Figure 5: $\frac{d}{dx}\tanh(x)$ (left) and $\frac{d}{dx}\text{sigmoid}(x)$ (right) have similar monotonicity and convexity/concavity properties.

**The segment to be bounded is concave if $-z_\sigma \leqslant l \leqslant u \leqslant z_\sigma$.** In this case, the lower bound is chosen to be the line directly connecting two endpoints $(l, \sigma(l))$ and $(u, \sigma(u))$. For the upper bound, $d^U$ can be any point between $l$ and $u$ (e.g. the middle point). See **(a)** of figure 6 (a).

**The segment to be bounded is convex if $z_\sigma \leqslant l \leqslant u$.** In this case, the upper bound is chosen to be the line directly connecting two endpoints. For the lower bound, $d^L$ can be any point between $l$ and $u$ (e.g. the middle point). See figure 6 (b).

**The segment is bounded similarly to an "S-shaped" function if $l < z_\sigma < u$.** See figure 6 (c).

- For the lower bound, let $d_l$ be the tangent point of the tangent line that passes through $(l, \sigma(l))$. If $u \leqslant d_l$, the lower bound is chosen to be the line directly connecting two endpoints. Otherwise, $d^L$ can be any point between $d_l$ and $u$ (e.g. the middle point).

- For the upper bound, let $d_u$ be the tangent point of the tangent line that passes through $(u, \sigma(u))$. If $l \geqslant d_u$, the upper bound is chosen to be the line directly connecting two

endpoints. Otherwise, $d^U$ can be any point between 0 and $d_u$ (e.g. linearly interpolated given $l$ ranging from $-u$ to $d_u$).

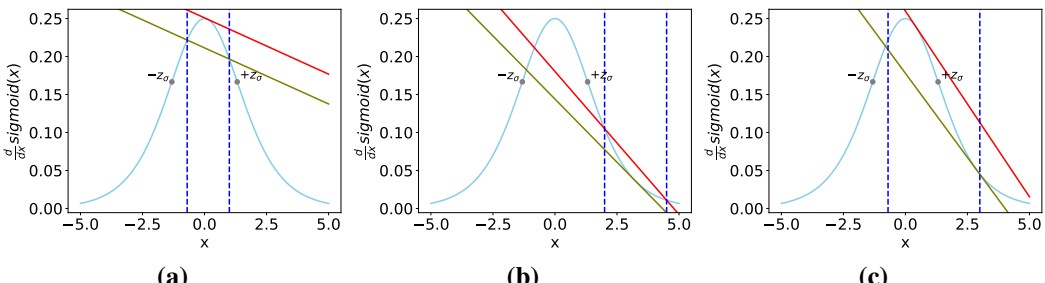

(a)                              (b)                              (c)

Figure 6: Illustration of linear relaxations of a "bell-shaped" function (sigmoid). **(a)** $-z_\sigma \leqslant l \leqslant u \leqslant z_\sigma$. The segment to be bounded is concave. **(b)** $z_\sigma \leqslant l \leqslant u$. The segment to be bounded is convex. **(c)** $l < z_\sigma < u$. The segment is bounded similarly to an "S-shaped" function.

## B.4 Verification Algorithm

Although a CEGIS procedure has been applied during training, small counterexamples can still exist. Previous works use bisection to identify the optimal boundaries of the region where the Lyapunov condition can be verified. This requires test running the entire verification procedure every time a new candidate boundary is set, which is very inefficient. In contrast, our work, thanks to the formulation of the region to verify as a levelset defined by $c_1$ and $c_2$ in 19, and the input-split branch-and-bound pipeline, can adaptively adjust the values of $c_1$ and $c_2$ *in a single verification run*, avoiding the tedious and time-consuming bisection process.

Specifically, during the branch-and-bound procedure, for each batch of subdomains, apart from identifying each of them as VERIFIED or UNKNOWN based on its certified bounds, we additionally empirically search for counterexamples on some of them (usually the one with the worst certified bound). We shall note that searching for counterexamples in a subdomain by either random or PGD attack is much easier than doing so in the whole initial domain, and thus is more likely to find counterexamples that were missed out in the CEGIS based training stage. We then shrink the levelset by increasing $c_1$ or decreasing $c_2$ to exclude all the counterexamples. The key intuition here is although the specification to verify is updated, the subdomains that has been verified remain verified, so we don't need to rerun the whole algorithm from the beginning again. The full verification algorithm is shown in algorithm 3.

## C Experiment Details

### C.1 Continuous Time Baselines

In the main text, we discussed the limitations of the continuous-time baselines [4, 12, 42]. A critical condition for theoretical guarantees is that the controller satisfies $u(x^*) = u^*$; otherwise, $x^*$ is no longer an equilibrium point, rendering the Lyapunov-based guarantee vacuous. In NLC [4], we observed that the provided code mistakenly uses a bias-free controller for both the inverted pendulum and the path-following dynamics. While this is correct for the inverted pendulum, where $u^* = 0$, it is incorrect for the path-following dynamics, as the chosen parametrization still enforces $u(0) = 0$, which does not hold in this case. This exact issue has also been noted in an open issue on NLC's official GitHub repository: `https://github.com/YaChienChang/Neural-Lyapunov-Control/issues/14`. For the method proposed by Wang et al. [42], we found no special treatment to enforce this condition. In fact, when testing the publicly released checkpoints from their GitHub repository, we found that none of them satisfied $u(x^*) = u^*$. For example, in the Double Integrator system, $u(0) = 0.3733$, and in the Van der Pol system, $u(0) = -0.0633$, despite both systems having $u^* = 0$ as their correct equilibrium input. Moreover, we believe there are some subtle issues in the verification formulation of Fossil 2.0. In [12], which underpins Fossil 2.0, we notice that their Certificate 2 misses the boundary condition, which should supposedly ensure either $\{x : V(x) \leqslant \beta\} \cap \mathcal{X}$ is empty or any trajectory starting from this intersection will remain in $\mathcal{X}$. We have observed that their method can

**Algorithm 3** Adaptive Branch-and-Bound Verification with Levelset Adjustment

---

**Require:** Neural network $f$, specification functions $V$ and $\dot{V} = \nabla V \cdot f$, initial domain $\Omega$, initial levelset thresholds $c_1 < c_2$, batch size $B$, boundary update margin $\epsilon$

 1: $\mathcal{S} \leftarrow \text{stack}([\Omega])$        $\triangleright$ stack of unverified subdomains
 2: **while** $\mathcal{S} \neq \varnothing$ **and** $c_1 < c_2$ **do**
 3:      $\mathcal{B} \leftarrow \text{POPBATCH}(\mathcal{S}, B)$        $\triangleright$ pop up to $B$ subdomains
 4:      $\mathcal{U} \leftarrow \varnothing$        $\triangleright$ unverified subdomains in this batch
 5:      **for** each $D$ in $\mathcal{B}$ **do**
 6:          Compute certified bounds of $V(x)$ and $\dot{V}(x)$ over $D$
 7:          **if** condition 19 is *not* verified to satisfy **then**      $\triangleright$ $c_1$ and $c_2$ are used here
 8:             add $D$ to $\mathcal{U}$
 9:      **if** $\mathcal{U} \neq \varnothing$ **then**
10:          $D_{\text{worst}} \leftarrow \text{WORSTDOMAIN}(\mathcal{U})$      $\triangleright$ greatest positive upper bound of $\dot{V}(x)$
11:          Empirically search $D_{\text{worst}}$ for counterexample $x_{\text{ce}}$
12:          **if** counterexample found **then**
13:             $\text{UPDATETHRESHOLDS}(x_{\text{ce}}, c_1, c_2, \epsilon)$
14:          **for** each $D$ in $\mathcal{U}$ **do**
15:             Split $D$ into $D_1, D_2$; $\text{PUSH}(\mathcal{S}, D_1, D_2)$        $\triangleright$ branching
16: **if** $\mathcal{S} = \varnothing$ **then**
17:      **return** VERIFIED        $\triangleright$ all subdomains have been verified
18: **else**
19:      **return** FALSIFIED        $\triangleright$ $c_1 = c_2$, the levelset is completely excluded

20: **function** $\text{UPDATETHRESHOLDS}(x_{\text{ce}}, c_1, c_2, \epsilon)$
21:      $v \leftarrow V(x_{\text{ce}})$
22:      **if** $|v - c_1| < |v - c_2|$ **then**
23:          $c_1 \leftarrow v + \epsilon$        $\triangleright$ counterexample near lower bound
24:      **else**
25:          $c_2 \leftarrow v - \epsilon$        $\triangleright$ counterexample near upper bound

---

occasionally return false positives, i.e., a controller and Lyapunov function are claimed to be found while the controller cannot stabilize any trajectory. Therefore, on top of their returned results, we ran an additional check with trajectory simulations, and recorded a negative for a run if the controller does not work.

## C.2 Dynamical Systems

**Van Der Pol:** The dynamics is given by

$$\dot{x}_1 = x_2$$
$$\dot{x}_2 = -x_1 + \mu(1 - x_1^2)x_2 + u$$

with $\mu = 1$. We choose the control input limit as $|u| \leqslant 1$ for all the baselines.

**Double Integrator:** The dynamics is given by

$$\dot{x}_1 = x_2$$
$$\dot{x}_2 = u$$

and constrain the torque limit as $|u| \leqslant 1$.

**Inverted Pendulum:** The dynamics is given as

$$\dot{\theta}_1 = \theta_2$$
$$\dot{\theta}_2 - \frac{\beta}{ml^2}\theta_2 + \frac{g}{l}\sin(\theta_1) + \frac{1}{ml^2}u.$$

We consider two set of torque limits as in [35]. The large torque case sets $|u| \leqslant 8.15mgl$ and the small torque case presents the more challenging limit $|u| \leqslant 1.02mgl$.

**Path Tracking:** The dynamics is given as

$$\dot{d}_e = v \sin(\theta_e)$$

$$\dot{\theta}_e = \frac{v}{l} u - \frac{\cos(\theta_e)}{\frac{r}{v} - \sin(\theta_e)}.$$

We also consider two set of torque limits for this dynamics. We set $|u| \leqslant 1.68\frac{l}{v}$ in the large torque case and $|u| \leqslant \frac{l}{v}$ in the small torque case following [35].

**Cartpole:** The dynamics is given by

$$\ddot{x} = \frac{1}{m_c + m_p \sin^2 \theta}(u + m_p \sin \theta (l\dot{\theta}^2 - g \cos \theta))$$

$$\ddot{\theta} = \frac{1}{l(m_c + m_p \sin^2 \theta)}(-u \cos \theta - m_p l \dot{\theta}^2 \cos \theta \sin \theta + (m_c + m_p)g \sin \theta).$$

Following DITL [44], we set $m_c = 1.0, m_p = 0.1, l = 1.0, g = 9.81$ and $|u| \leqslant 30$.

**2D Quadrotor:** The dynamics is given by

$$\ddot{x} = -\frac{1}{m} \sin(\theta)(u_1 + u_2)$$

$$\ddot{y} = \frac{1}{m} \cos(\theta)(u_1 + u_2) - g$$

$$\ddot{\theta} = \frac{l}{I}(u_1 - u_2)$$

with $m = 0.486, l = 0.25, I = 0.00383, g = 9.81$. We also set the control input $u \in \mathbb{R}^2_{\geqslant 0}$ to have the constraints $\|u\|_\infty \leqslant 1.25mg$ as in [35].

**PVTOL:** The dynamics is given by

$$\dot{x}(t) = \begin{bmatrix} v_x \cos\phi - v_z \sin\phi \\ v_x \sin\phi + v_z \cos\phi \\ \dot{\phi} \\ v_z \dot{\phi} - g \sin\phi \\ -v_x \dot{\phi} - g \cos\phi \\ 0 \end{bmatrix} + \begin{bmatrix} 0 & 0 \\ 0 & 0 \\ 0 & 0 \\ 0 & 0 \\ \frac{1}{m} & \frac{1}{m} \\ \frac{l}{J} & -\frac{l}{J} \end{bmatrix} u \tag{27}$$

where $g = 9.8, m = 4.0, l = 0.25, J = 0.0475$. We set the control input $u \in \mathbb{R}^2_{\geqslant 0}$ to have the limit $\|u\|_\infty \leqslant 39.2$ as in [44].

**Ducted Fan:** The dynamics is given by

$$\ddot{x} = \frac{1}{m}(-d\dot{x} + u_0 \cos\theta - u_1 \sin\theta)$$

$$\ddot{y} = \frac{1}{m}(-d\dot{y} + u_0 \sin\theta + u_1 \cos\theta) - g$$

$$\ddot{\theta} = \frac{r}{I} u_0$$

with $m = 11.2, r = 0.156, I = 0.0462, d = 0.1$, and $g = 0.28$. We set the control input $u = [u_0, u_1]^\top \in \mathbb{R}^2$ to be $u_0 \in [-10, 10], u_1 \in [0, 10]$.

**3D Quadrotor:**

$$\ddot{\mathbf{p}} = \begin{bmatrix} 0 \\ 0 \\ -g \end{bmatrix} + \frac{1}{m} R_{\text{col}}(\phi, \theta, \psi) \cdot T$$

$$\dot{\phi} = \omega_x + \tan(\theta)(\sin(\phi)\omega_y + \cos(\phi)\omega_z)$$

$$\dot{\theta} = \cos(\phi)\omega_y - \sin(\phi)\omega_z$$

$$\dot{\psi} = \frac{\sin(\phi)\omega_y + \cos(\phi)\omega_z}{\cos(\theta)}$$

$$\dot{\boldsymbol{\omega}} = \mathbf{I}^{-1}(-\boldsymbol{\omega} \times (\mathbf{I}\boldsymbol{\omega}) + \boldsymbol{\tau})$$

where

$$R_{\text{col}}(\phi, \theta, \psi) = \begin{bmatrix} \sin\psi\sin\phi + \cos\psi\sin\theta\cos\phi \\ -\cos\psi\sin\phi + \sin\psi\sin\theta\cos\phi \\ \cos\theta\cos\phi \end{bmatrix}$$

The control input $u \in \mathbb{R}^4$ represents the four rotor thrusts. These are mapped to generalized forces via

$$\begin{bmatrix} T \\ \tau_x \\ \tau_y \\ \tau_z \end{bmatrix} = \begin{bmatrix} 1 & 1 & 1 & 1 \\ 0 & l & 0 & -l \\ -l & 0 & l & 0 \\ c & -c & c & -c \end{bmatrix} u$$

The hyperparameters are specified with $m = 0.486, l = 0.225, g = 9.81, \mathbf{I} = \text{diag}(I_x, I_y, I_z) = \text{diag}(0.0049, 0.0049, 0.0088), c = \frac{1.1}{29} \approx 0.03793, u\text{*} = \frac{mg}{4} = \frac{0.486 \times 9.81}{4} = 1.190325$. The torque limit of $u \in \mathbb{R}^4_{\geq 0}$ is given by $|u| \leqslant 3.6$.

### C.3 Examples of Domain Update

We present some examples on the effect of the dynamic domain expansion mechanism and demonstrate its effectiveness. The initial domain for 2D Quadrotor is picked by 0.4 multiplied the original domain from [48, 35], and the other ones are designed to be small but overall are picked quite arbitrarily without much tuning. A comparison between the initial training domain and the one after the first stage training is presented in Table 4. We can notice that our domain update mechanism is able to enlarge the training domain and thus the ROA by a large amount, resulting in much bigger and much more less conservative ROA estimation compared to the baselines.

Table 4: Example start and end domain for each dynamical system. Lower/Upper bound of the domain is given.

| System | Start Domain | End Domain |
|---|---|---|
| Double Integrator | $\pm[1, 1]$ | $\pm[27.6, 9.6]$ |
| Van der Pol | $\pm[1, 1]$ | $\pm[6, 8.5]$ |
| Pendulum (small torque) | $\pm[1, 2]$ | $\pm[27.6, 69.6]$ |
| Pendulum (large torque) | $\pm[1, 2]$ | $\pm[30, 120]$ |
| Path Tracking (small torque) | $\pm[2, 2]$ | $\pm[9.6, 7.2]$ |
| Path Tracking (large torque) | $\pm[2, 2]$ | $\pm[12, 12]$ |
| Cartpole | $\pm[0.4, 0.4, 0.4, 0.4]$ | $\pm[4.8, 3.6, 13.2, 13.2]$ |
| 2D Quadrotor | $\pm[0.3, 0.3, 0.2\pi, 1.6, 1.6, 1.2]$ | $\pm[12, 13.2, 12, 19.2, 20.4, 88.8]$ |
| PVTOL | $\pm[0.4, 0.4, 0.4, 0.4, 0.4, 0.4]$ | $\pm[3.6, 3.6, 3.6, 6, 8.4, 33.6]$ |
| Ducted Fan | $\pm[0.4, 0.4, 0.4, 0.4, 0.4, 0.4]$ | $\pm[6, 7.2, 10.8, 4.8, 3.6, 25.2]$ |

### C.4 Examples of verifiable $c_1$ and $c_2$

Table 5 gives some examples of the final verifiable $c_1, c_2$ thresholds. To demonstrate that the unverifiable hole is small, we also report the volume of the sublevelset $V^{\leqslant c_1}$ and the volume it occupies the full ROA. We can see that for all the systems the unverifiable region occupies only less than $0.1\%$ of the full ROA. For the higher-dimensional (6-D) systems, none of the $5 \times 10^7$ samples fell into the small unverifiable region.

### C.5 Training Time

Table 6 provides the training time for our method. The time needed for the ROA estimation stage is relatively stable since it is trained to a fixed epoch. However, the time taken by CEGIS will depend on the $c_1$ chosen and the hardness of counterexamples near $c_1$. In practice, we found that even though the CEGIS time varies, all training processes complete within reasonable time.

### C.6 More ROA Visualizations

We also provide more visualizations on the final ROA obtained through our methods in figure 7. Across all visualized slices, our method yield much larger ROA estimation compared to the baselines.

Table 5: Examples of verifiable thresholds $c_1$, $c_2$, and the volume of the unverifiable sublevel set $\{x : V(x) \leqslant c_1\}$ with its proportion of the full ROA. The ROA is estimated in a Monte-Carlo fashion with 50000000 points.

| System | $c_1$ | $c_2$ | Vol$\{V(x) \leqslant c_1\}$ | Proportion to Full ROA |
|--------|-------|-------|-------------------|------------------------|
| Double Integrator | 0.01 | 0.99 | 0.038 | 0.014% |
| Van der Pol | 0.0106 | 0.99 | 0.015 | 0.025% |
| Pendulum (small torque) | 0.0103 | 0.98 | 0.17 | 0.016% |
| Pendulum (large torque) | 0.022 | 0.98 | 0.36 | 0.012% |
| Path Tracking (small torque) | 0.0108 | 0.99 | 0.057 | 0.08% |
| Path Tracking (large torque) | 0.0105 | 0.99 | 0.040 | 0.03% |
| Cartpole | 0.0068 | 0.95 | 0.0154 | 0.003% |
| 2D Quadrotor | 0.096 | 0.89 | 0 | 0% |
| PVTOL | 0.003 | 0.90 | 0 | 0% |
| Ducted Fan | 0.0058 | 0.89 | 0 | 0% |

Table 6: Training time (in seconds) for each system. We report separately for the ROA Estimation stage, CEGIS stage, and a combined overall time in seconds.

| System | ROA Estimation | CEGIS | Full |
|--------|----------------|-------|------|
| Double Integrator | $1173.20 \pm 30.83$ | $162.39 \pm 97.61$ | $1335.6 \pm 123.80$ |
| Van der Pol | $1201.10 \pm 25.35$ | $187.53 \pm 28.14$ | $1388.64 \pm 44.38$ |
| Pendulum (small torque) | $1169.27 \pm 139.29$ | $225.14 \pm 64.17$ | $1394.41 \pm 134.53$ |
| Pendulum (large torque) | $2946.59 \pm 166.73$ | $98.25 \pm 44.97$ | $3044.85 \pm 153.15$ |
| Path Tracking (small torque) | $789.30 \pm 29.64$ | $131.55 \pm 54.61$ | $920.85 \pm 62.67$ |
| Path Tracking (large torque) | $820.67 \pm 42.41$ | $161.67 \pm 48.19$ | $982.35 \pm 81.07$ |
| Cartpole | $4421.06 \pm 78.97$ | $3182.58 \pm 879.25$ | $7603.65 \pm 881.36$ |
| 2D Quadrotor | $5704.92 \pm 251.83$ | $12050.59 \pm 7258.85$ | $17755.52 \pm 7110.40$ |
| PVTOL | $7100.86 \pm 192.40$ | $10240.46 \pm 1554.67$ | $17341.33 \pm 1570.56$ |
| Ducted Fan | $6132.89 \pm 408.35$ | $12129.13 \pm 3419.57$ | $18262.03 \pm 3187.87$ |
| 3D Quadrotor | $4721.13 \pm 306.55$ | – | $4721.13 \pm 306.55$ |

## C.7 Computation Resource

The training was performed on NVIDIA V100 gpus, each with 16GB memory. The verification was performed on NVIDIA 5090 gpus with 32GB memory.

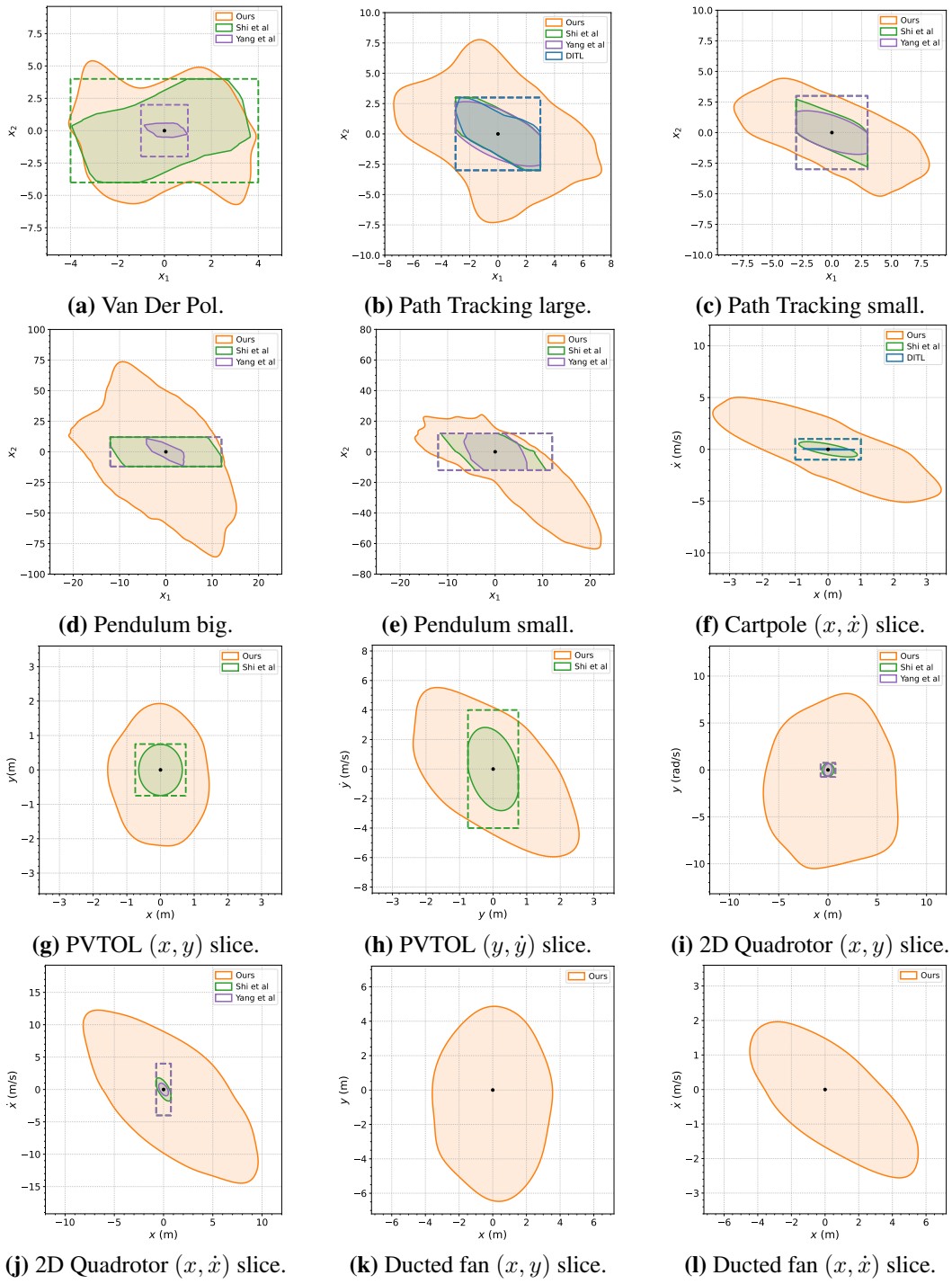

Figure 7: More ROA visualizations.

