# OpenReview forum: "Two‑Stage Learning of Stabilizing Neural Controllers via Zubov Sampling and Iterative Domain Expansion"
_NeurIPS.cc/2025/Conference — NeurIPS 2025 spotlight_

### Official Review · Reviewer_NRaM · 2025-06-15

**Clarity:** 3
**Significance:** 3
**Originality:** 3
**Rating:** 4
**Confidence:** 2

**Summary:**

This paper proposes a new two-stage Zubov-based training framework for synthesizing stabilizing neural network controllers, it contains a curriculum training domain expansion and training sample selection that significantly reduces the conservatism during training. The author also improves the advanced neural network verification tool and propose a novel verification algorithm. The empirical results demonstrate large improvements in ROA estimation and achieves significant verification speedups.

**Questions:**

- The paper overall looks good to me, but I have a general question reguarding to this paper -- why this paper is send to NeurIps rather than ICCAD or DAC? What's it contribution reguarding to ML side?

**Ethical Concerns:**

["NO or VERY MINOR ethics concerns only"]

**Final Justification:**

I have read the author's response and it has resolved all my mentioned concerns. So I keep the score as borderline accepted.

**Limitations:**

Yes

**Quality:**

3

**Strengths And Weaknesses:**

Strength

- The paper is logically organized and clearly written. The problem setup/motivation/methods are introduced with appropriate theoretical grounding.

- The approach shows strong empirical scalability. And the speedups in verification and the increased ROA volumes have practical significance.

Weakness

- The lack of formal guarantees in higher-dimensional systems limits the strength of the paper's claims in safety-critical applications. The dependence on a verifier that still struggles with high-dimensional tight bounds means that many results are still empirical.

- This is more likely a control-theoretic contribution, its broader impact within ML remain to be discussed.

---

> ### Author Rebuttal · Authors · 2025-07-31
>
> We appreciate the reviewer’s positive feedback. Below, we address each question the reviewer raised for our work.
>
> ### **Weakness 1: Verification in High Dimensions**
> We agree with the reviewer that scaling verification to high-dimensional systems remains a key bottleneck, even though our approach achieves state-of-the-art performance. We want to mention that we are actively pushing the frontiers of this research direction and have demonstrated results that are unattainable with existing baselines. We would love to motivate more people to study this problem and further improve the verification pipeline for better real-world applications.
>
> ### **Weakness 2 & Question 1: Why NeurIPS and Significance to the ML community**
> Although ICCAD and DAC are great venues with papers on relevant topics, we believe our work presents its unique value to the machine learning community due to the following three reasons:
>
> * Our work lies at the **intersection of machine learning and control theory**, addressing the important problem of learning-based control. It demonstrates how machine learning techniques can be applied to a fundamental challenge in engineering and science. **Similar lines of work have been published in leading ML venues.**. For example, [1] is published at NeurIPS 2019, [2] at NeurIPS 2023, and [3] at ICML 2024.
> * More broadly, our work contributes to the problem of **Lyapunov function synthesis**, which has recently received increasing attention in the ML community, with related papers such as [4] at ICML 2025 and [5] at CVPR 2025. We believe our work shares similar merit and is suitable for publishing at a machine learning conference.
> * Finally, our work extends the **neural network verification** toolbox α,β-CROWN ([6-11, NeurIPS 2018, 2020, 2021, 2022, 2023, 2024]) in some nontrivial ways, such as **enabling automatic bound computation over Jacobian operators and providing new tight relaxations** for many commonly used derivatives.
>
> We believe that publishing this work at a comprehensive machine learning venue like NeurIPS will engage both the neural network verification and control communities, and help make progress on the broader challenge of scalable, verifiable learning based control.
>
> ### **References**
> [1] Chang, Ya-Chien, Nima Roohi, and Sicun Gao. "Neural Lyapunov control." _Advances in neural information processing systems_ 32 (2019).
> [2] Wu, Junlin, et al. "Neural Lyapunov control for discrete-time systems." _Advances in neural information processing systems_ 36 (2023): 2939-2955.
> [3] Yang, Lujie, et al. "Lyapunov-stable neural control for state and output feedback: A novel formulation." _arXiv preprint arXiv:2404.07956_ (2024).
> [4] Zou, Haohan, et al. "Analytical Lyapunov Function Discovery: An RL-based Generative Approach." _arXiv preprint arXiv:2502.02014_ (2025).
> [5] Zhao, Hanrui, et al. "Learning-enabled Polynomial Lyapunov Function Synthesis via High-Accuracy Counterexample-Guided Framework." _Proceedings of the Computer Vision and Pattern Recognition Conference_. 2025.
> [6] Zhang, Huan, et al. "Efficient neural network robustness certification with general activation functions." Advances in neural information processing systems 31 (2018).
> [7] Xu, Kaidi, et al. "Automatic perturbation analysis for scalable certified robustness and beyond." Advances in Neural Information Processing Systems 33 (2020): 1129-1141.
> [8] Wang, Shiqi, et al. "Beta-crown: Efficient bound propagation with per-neuron split constraints for neural network robustness verification." Advances in neural information processing systems 34 (2021): 29909-29921.
> [9] Zhang, Huan, et al. "General cutting planes for bound-propagation-based neural network verification." Advances in neural information processing systems 35 (2022): 1656-1670.
> [10] Kotha, Suhas, et al. "Provably bounding neural network preimages." Advances in Neural Information Processing Systems 36 (2023): 80270-80290.
> [11] Zhou, Duo, et al. "Scalable neural network verification with branch-and-bound inferred cutting planes." Advances in Neural Information Processing Systems 37 (2024): 29324-29353.

---

> > ### Comment · Reviewer_NRaM · 2025-08-01
> >
> > Thanks for the clarification! My concerns are well addressed.

---

> > > ### Author Response · Authors · 2025-08-04
> > >
> > > Thank you for your response! We are glad to hear that your concerns are well addressed. We are happy to take in any further questions you have regarding our work!

---

### Official Review · Reviewer_N7NM · 2025-07-01

**Clarity:** 3
**Significance:** 3
**Originality:** 3
**Rating:** 5
**Confidence:** 4

**Summary:**

This paper presents a novel two-stage learning framework for synthesizing stabilizing neural controllers with formal stability guarantees. The authors propose a Zubov-inspired sampling strategy and iterative domain expansion mechanism to significantly reduce conservatism in training neural Lyapunov functions, while extending the α,β-CROWN verifier to handle Jacobian-related operators for efficient formal verification. The method demonstrates remarkable improvements, achieving 5-1.5×10^5 times larger regions of attraction compared to baselines and 40-10000× faster verification than SMT solvers across various nonlinear systems from 2D to 12D. The work makes important contributions in bridging the gap between empirical neural control performance and formal stability guarantees through its innovative training and verification pipeline.

**Questions:**

How does the method's computational complexity scale with system dimensionality, particularly for the verification stage which still faces challenges in higher dimensions?

Could you elaborate on the robustness of the approach when dealing with model uncertainties or disturbances not accounted for during training?

What modifications would be needed to apply this framework to hybrid or discrete-time systems commonly encountered in robotics applications?

Have you considered combining the approach with online adaptation mechanisms to maintain stability guarantees under slowly varying system dynamics?

**Ethical Concerns:**

["NO or VERY MINOR ethics concerns only"]

**Final Justification:**

The proposed pipeline achieves impressive verified regions of attraction, yet its reliance on careful hyper-parameter tuning and its limited treatment of uncertainty, hybrid dynamics, and time-varying systems leave notable gaps.

**Limitations:**

Yes

**Quality:**

3

**Strengths And Weaknesses:**

While the methodology shows impressive empirical results across multiple nonlinear systems (achieving ROA volumes 5-1.5·10^5 times larger than baselines), its formal verification capabilities remain limited for higher-dimensional systems due to computational constraints in Jacobian-bound propagation. The theoretical contributions are well-grounded with proper assumptions and proofs, though the practical implementation requires careful hyperparameter tuning that may challenge real-world adoption.

---

> ### Author Rebuttal · Authors · 2025-07-31
>
> We thank the reviewer for acknowledging our contribution and for highlighting several promising directions for future work. Below, we will have a point-to-point response to each of the questions raised.
> ### **Weakness 1: Verification in High Dimension**
> We share the reviewer’s concern that scaling verification to high-dimensional systems remains a key bottleneck, even though our method currently sets the state-of-the-art. We want to mention that we are actively pushing the frontiers of this research direction and have demonstrated results that are not possible with existing baselines. We would love to motivate more people to study this problem and further improve the verification pipeline for better real-world applications.
>
> ### **Weakness 2: Hyperparameter Tuning Difficulty**
> We thank the reviewer for bringing this up. We do agree that learning a stabilizing controller is quite a challenging setting, and our method is working towards the SOTA on making the tuning easier and training more stable. We have simplified the two most difficult hyperparameters, training domains and initializations, to tune in baselines such as [2,3], thanks to our novel training pipeline, and we achieved a much better success rate in Table 1.
>
> * One difficult hyperparameter is the **training domain**; selecting it not optimally may cause complete failure of the training. However, we have made this **hyperparameter selection mostly automatic** with our novel dynamic domain expansion. As Table 4 in Appendix C.3 shows, we begin with a loosely chosen initial domain and rely on dynamic expansion to refine it on the fly, yet the method still achieves very strong performance.
> * Another difficult hyperparameter is the **initialization**. Previous work depends on RL/LQR initializations can fail easily with a badly chosen initialization parameter. However, using our novel two-stage training framework, we eliminate the need for special initializations and achieve strong performance by training from scratch.
>
> ### **Question 1: Verification Complexity**
> Unfortunately, there is no simple or uniform answer. In practice, verification complexity can **depend on many factors**, such as the system dimensions, the range of the verification domain, and the model architecture. For example, alpha,beta-CROWN was originally designed for robustness verification, where a typical problem is handling a perturbed image input through a CNN. In this setting, the input dimension is huge (e.g., 64 * 64 * 3); however, the verifier is still able to handle it due to the allowed perturbation on each dimension being small (around 0.01), and the computation graph is relatively simple. In our setting, each dimension of the **input domain can be huge** (around 10), and the **computation graph has a very complicated topology** due to the complexity of the dynamical system and the Lyapunov condition. Moreover, the computation graph involves **many unique nonlinear operators** such as tanh, sigmoid, their derivatives, and multiplication with both operands perturbed, which are hard to bound tightly. This issue presents a **common challenge** for all existing verification approaches to this problem. Despite this, **our training and verification method currently achieves state-of-the-art performance** for this class of problems, and we believe it advances the boundary of what is possible in high-dimensional nonlinear system verification.
>
> ### **Question 2: Robustness under uncertainties/disturbances**
> This is a very interesting direction that well deserves further exploration, although it falls outside the scope of our current paper. We believe verifying the trained model’s robustness under uncertainties or disturbances is **non-trivial**, as it would require a novel verification formulation of robust control theories like dissipativity theory. It is also possible that we need to reformulate our training framework under the robust control framework, which can be more non-trivial since it requires the combination of Zubov theory and dissipativity theory.
>
> ### **Question 3: Extension to discrete/hybrid systems**
> Our current training framework’s success depends on the existence of a Zubov-type theory that can characterize the underlying ROA. For discrete-time systems, we notice that there is some recent work that discusses the discrete-time Zubov theorem [1], and our work should be readily **extendable to discrete-time settings**. However, there are still some **non-trivial challenges** in the application of dynamic domain expansion due to some connectedness issues. It is also highly non-trivial to extend our approach to hybrid systems, since a Zubov-type theory can be **more difficult to obtain** in this setting, and would require novel combinations of Zubov theory, hybrid system theory, and nonsmooth analysis. Both of these settings are beyond the scope of our current paper, but we believe it is an interesting and promising future direction.
>
> ### **Question 4: Extension to slowly time-varying systems**
> Combining our approach with online adaptation mechanisms to maintain stability guarantees under slowly varying system dynamics is also not trivial. With prescribed bounds on the slow-changing parameter and its changing rate, we believe it is possible to address the slowly varying parameter in the robust control framework and combine our approach with dissipativity theory to train a robust controller. However, as commented above, combining our approach with dissipativity theory is highly non-trivial in the first place, and hence, **more future work will be needed** to pursue this interesting direction.
>
> ### **References**
> [1] Serry, Mohamed, et al. "Safe Domains of Attraction for Discrete-Time Nonlinear Systems: Characterization and Verifiable Neural Network Estimation." _arXiv preprint arXiv:2506.13961_ (2025).
> [2] Yang, Lujie, et al. "Lyapunov-stable neural control for state and output feedback: A novel formulation." arXiv preprint arXiv:2404.07956 (2024).
> [3] Shi, Zhouxing, Cho-Jui Hsieh, and Huan Zhang. "Certified training with branch-and-bound: A case study on lyapunov-stable neural control." arXiv preprint arXiv:2411.18235 (2024).

---

> > ### Comment · Reviewer_N7NM · 2025-08-07
> >
> > Thank you for the detailed responses. I appreciate the clarifications regarding hyper-parameter tuning and the acknowledgement that robustness to uncertainty, discrete/hybrid systems, and time-varying dynamics remain open challenges. Please ensure these limitations are explicitly discussed in the revised manuscript.

---

> > > ### Author Response · Authors · 2025-08-07
> > >
> > > Thank you for your response. We will make sure to discuss these limitations and clarify that they are still open challenges in our revised paper. We would be happy to address any further questions you have regarding our work!

---

### Official Review · Reviewer_QzKe · 2025-07-02

**Clarity:** 3
**Significance:** 3
**Originality:** 3
**Rating:** 5
**Confidence:** 3

**Summary:**

This paper presents a novel approach for learning stabilizing neural controllers and corresponding Lyapunov functions, grounded in Zubov’s partial differential equation (PDE) framework. The motivation for using Zubov’s theorem lies in its natural connection to estimating the region of attraction (ROA). The authors introduce a Zubov-inspired sampling strategy that combines points from both inside and outside the current ROA estimate, allowing for more informed learning. This is followed by a counterexample-guided inductive synthesis (CEGIS) loop that iteratively refines the candidate Lyapunov function until a verifiable certificate is obtained. The framework employs the
alpha-beta-CROWN verifier for neural network verification and is evaluated across a diverse set of benchmarks, from simple 2D systems like the inverted pendulum to a 12-dimensional 3D quadrotor. Experimental results demonstrate that the proposed approach produces significantly larger ROA estimates compared to baselines and achieves superior scalability in neural network verification relative to dReal.

**Questions:**

The FOSSIL framework appears to be a state-of-the-art tool for synthesizing neural Lyapunov certificates. How does your approach compare with FOSSIL in terms of ROA estimation, verification strength, and scalability? Could you clarify the reason for not including a direct comparison?

**Ethical Concerns:**

["NO or VERY MINOR ethics concerns only"]

**Final Justification:**

The authors have adequately addressed my concerns. I will maintain my original positive score.

**Limitations:**

As noted earlier, the paper does not include comparisons with closely related tools like FOSSIL.

**Paper Formatting Concerns:**

Technical terms such as "forward invariants" are used without explicit definition.

**Quality:**

4

**Strengths And Weaknesses:**

Strengths

1. The paper studies a timely and well-motivated problem. It is well-written and technically sound, with a non-trivial and original approach.
2. The experimental benchmarks span a wide range of complexity and dimensionality, clearly demonstrating the benefits of the proposed approach relative to existing baselines.
3. The authors implement several practical enhancements to alpha-beta CROWN, including linear relaxations of Jacobian operators and a branch-and-bound heuristic, improving the scalability and efficiency of verification.

Weaknesses
1. The introduction would benefit from a clearer articulation of the novelty of leveraging Zubov’s theorem in contrast to more traditional Lyapunov-based approaches.
2. The paper omits several relevant citations related to CEGIS-based synthesis of Lyapunov functions, including:
- Formal Synthesis of Lyapunov Neural Networks, Abate et al.
- Counterexample-Guided Inductive Synthesis of Control Lyapunov Functions for Uncertain Systems, Masti et al., IEEE
- Counterexample-Guided Synthesis of Control Lyapunov Functions for Switched Systems, Ravanbakhsh and Sankaranarayanan, IEEE
- FOSSIL: A Software Tool for the Formal Synthesis of Lyapunov Functions and Barrier Certificates Using Neural Networks, Abate et al., ACM

---

> ### Author Rebuttal · Authors · 2025-07-31
>
> We thank the reviewer for recognizing our contribution. We here provide a point-to-point response to the reviewer's question.
>
> ### **Weakness 1: More introduction on Zubov approach**
> Thank you for your suggestion, we will add a more detailed explanation of Zubov theorem in the introduction to clearly distinguish our work from the works that only utilize traditional Lyapunov theory.
>
> ### **Weakness 2 & Question 1: Fossil baseline and discussion**
> Thank you for pointing out the missing citations and baselines. We will add the citations of the mentioned papers, and we have since compared our method against Fossil 2.0 and present the results here.
>
> We notice that Fossil has the same issue as NLC mentioned in Appendix C.1, as it **falsely assumes that $u(0) = 0$ on the equilibrium**. Thus, we here only present their results on the systems that satisfy the property $u(0) = 0$. It can be noticed that it **does not scale to the 4D** cartpole system, and **achieves smaller ROA volumes** than our method in 2D systems.
>
> |Systems|Fossil 2.0||Ours||
> |:-:|:-:|:-:|:-:|:-:|
> ||ROA|Succ|ROA|Succ|
> |Van Der Pol|3.05±0.58|60%|**57.6±3.4**|100%|
> | Double Int|258.1±36.3|80%|**302.5±10.7**|100%|
> | Pendulum S|106.5 ± 23.5|80%|**1169.2±124.5**|100%|
> | Pendulum B|285.2±53.3|80%|**2946.5±149.1**|100%|
> | Cartpole|—|0%|**306.1±54.2**|100%|
>
> Regarding scalability and verification strength, Fossil is more similar to NLC [1] and uses SMT solvers like dReal or Z3 for verification and counterexample generation. Since we are using the more scalable and powerful verifier α,β-CROWN and PGD-based quick counterexample sampling, **our method is better in both verification and training scalability**, which is also proved by the experimental results.
>
> ***Side note about Fossil.***
> We believe there are some subtle issues in their verification formulation. In reference [2], which underpins Fossil 2.0, we notice that their Certificate 2 **misses the boundary condition**, which should supposedly ensure either $\lbrace x: V(x) \leq \beta \rbrace \cap \mathcal{X}$ is empty or any trajectory starting from this intersection will remain in $\mathcal{X}$. We have observed that their method can occasionally return false positives even though $\mathcal{X}$ is already chosen to be much bigger than $\mathcal{X}_I$, i.e., a controller and Lyapunov function are claimed to be found while the controller cannot stabilize any trajectory. Therefore, on top of their returned results, we ran an additional check with trajectory simulations, and recorded a negative for a run if the controller does not work.
>
> ### **References**
> [1] Chang, Ya-Chien, Nima Roohi, and Sicun Gao. "Neural lyapunov control." _Advances in neural information processing systems_ 32 (2019).
> [2] Edwards, Alec, Andrea Peruffo, and Alessandro Abate. "A general framework for verification and control of dynamical models via certificate synthesis." _arXiv preprint arXiv:2309.06090_ (2023).

---

> > ### Comment · Reviewer_QzKe · 2025-08-05
> >
> > Thank you for your detailed response. You have addressed all of my concerns.

---

> > > ### Author Response · Authors · 2025-08-05
> > >
> > > We appreciate your response and are glad your concerns have been clarified. We are happy to address any potential further questions you have regarding our work!

---

### Official Review · Reviewer_Pjvg · 2025-07-03

**Clarity:** 2
**Significance:** 3
**Originality:** 2
**Rating:** 4
**Confidence:** 4

**Summary:**

This paper considers the problem of neural control for stabilization tasks with formal guarantees on correctness. It proposes a method for jointly synthesizing a neural control policy and a neural Lyapunov function, where the goal is also to maximize the region of attraction (ROA) within which the resulting neural control policy is guaranteed to stabilize the system. Following several recent works, the present method follows a counterexample-guided inductive synthesis (CEGIS) based approach where a learner and a verifier module are composed in a loop. The novelty behind the present method is that it proposes a novel loss function for the learning module, which is inspired by Zubov's theorem characterization of the ROA. This loss function allows more explicit maximization of the ROA compared to prior methods. On the verification side, the method utilizes three verification schemes. The first verification scheme is a formal verification method that uses interval bound propagation combined with linear relaxations of involved non-linear functions which can then be achieved using $\alpha$-$\beta$-CROWN. The second and the third verification scheme are empirical in nature and do not provide formal guarantees. The method is implemented and experimentally evaluated on a number of benchmarks collected from the literature, and the experiments show that the method is able to learn neural control policies and neural Lyapunov functions that give rise to larger ROAs compared to the baselines.

**Questions:**

1. Questions about the verification algorithm (see Weakness 2 above): To what function exactly is interval bound propagation applied? How is this achieved when the dynamics function $f$ is non-linear and involves non-linearities beyond sigmoid or tanh functions?

2. In the experiments, which benchmarks have been successfully verified by the first verification scheme that provides formal guarantees?

**Ethical Concerns:**

["NO or VERY MINOR ethics concerns only"]

**Final Justification:**

The author response addresses and clarifies all my questions and I do feel more positive about the paper now, especially with the question about which experiments have been formally verified being clarified. I have increased my score accordingly.

**Limitations:**

The paper should be precise about which benchmarks in the experiments were formally verified, given that formal guarantees constitute one of the main motivations behind the paper.

**Paper Formatting Concerns:**

I did not observe any formatting issues.

**Quality:**

2

**Strengths And Weaknesses:**

**Strengths**

1. *Problem relevance and novelty.* The paper considers an important problem of neural control policy synthesis with formal stabilization guarantees. While this problem has received considerable attention, the proposed loss function that is motivated by Zubov's theorem is a new idea that more explicitly reasons about the ROA, hence allowing its maximization. This is a novel feature that distinguishes the present method from most previous works on this topic.

2. *Experiments (learning).* Experiments demonstrate that the method is able to learn neural control policies and neural Lyapunov functions that provide larger ROAs compared to the baselines. It should be noted that, apart from lower dimensional benchmarks, these ROAs are not formally verified (see weaknesses below).

**Weaknesses**

1. *Presentation and the lack of clarity.* The technical sections of the paper are very hard to follow and only provide a high level overview. To actually understand any details of the learning procedure one needs to read the appendix, and even after reading the paper + appendix there are several aspects of the method that are unclear to me. This makes the paper non-self contained and ambiguous, and I believe that the presentation requires a major revision for the paper to be publishable : \
** For selecting training points inside and outside ROA, in l.159-165 it is proposed to take points with V^{<=c} as inside ROA and points at which V is close to 1 as outside ROA. But how is the value of this threshold c determined? Also, wouldn't this result in points outside ROA being very far away from the ROA boundary? \
** The loss function used for training is never properly defined in the paper and is deferred to the appendix. In l.181, it is simply said that the loss function is easily adapted from Zubov's theorem. This is insufficient for a paper to be self contained. While I eventually understood the loss function, it took me a while to decipher it since the appendix provides no intuition on how each loss term corresponds to conditions in Theorem 2.3 and 2.4. For instance, it is never properly mentioned that the positive definite function $\omega$ in Theorem 2.4 is set to $\omega(x) = ||x||^p$. \
** I could not understand the part on the CEGIS loop in Section 3.1.2 at all, especially how counterexamples are incorporated into the loss function, even after several reads.

2. *Ambiguity about the verification procedure.* The verification procedure in Section 3.2 is described by basically saying that the method uses interval bound propagation and linear relaxation of non-linear functions such as sigmoid or tanh, which can then be delegated to $\alpha$-$\beta$-CROWN. But no further details are provided. To what function exactly is interval bound propagation applied? How is this achieved when the dynamics function $f$ is non-linear and involves non-linearities beyond sigmoid or tanh functions? At this point, I cannot say that I understand the verification procedure at all.

3. *No guarantees around the origin.* The verification method does not provide formal guarantees on stability around the origin, see l.218-222. This is a disadvantage compared to some SMT-based methods that do address this issue but are not discussed in the paper, e.g. \
[1] Abate et al. "Formal Synthesis of Lyapunov Neural Networks". IEEE Control Systems Letters 5 (3), 2020

4. *Experiments not formally verified for larger benchmarks.* The paper is motivated by the need for formal guarantees and achieving them without reliance on SMT solvers. However, when it comes to designing the verification module, the paper considers three verification schemes of which only the first one provides formal guarantees. The second and the third verification scheme are empirical and do not actually provide guarantees. While Section 4 is ambiguous about which verification scheme was applied to which benchmark, from the text my understanding is that only 2D and 4D benchmarks were verified by the first verification scheme. But this is in line with what existing SMT-based methods can formally verify as well, e.g. reference [8] in the paper which is not considered in the experiments. This should be properly clarified, also in the abstract and into.

**Minor comments**
- l.29 searching the Lyapunov function -> searching for the Lyapunov function
- l.75 the the -> the
- l.100 levelset -> level set
- l.110 function satisfies -> function that satisfies
- l.117 can obtained -> can be obtained
- eq. (14) and l.255 is $g$ supposed to be $f$?

---

> ### Author Rebuttal · Authors · 2025-07-31
>
> We appreciate the reviewer’s careful reading, which uncovered several places where our exposition is unclear. While the current draft could undoubtedly be clearer, we believe the main concerns arise from presentation gaps rather than from substantive flaws. Below, we address each point to clarify misunderstandings, and we promise that these clarifications will appear in the revision. In addition, we also conducted **additional experiments** to compare with a new SMT baseline, fossil.
>
> ### **W1: Clarity on training**
>
> ***Value c & Outside ROA data selection.*** In our real experiments, this $c$ can be chosen to be 0.99 for all 2D systems and needs to be slightly more conservative for higher-dimensional systems (0.95 for 4D and 0.9 for even higher-dimensional systems). For the selection of data points outside the ROA, we shall notice that they are still constrained **within the training domain** and cannot diverge to infinity since we adopted **projected gradient descent**. Moreover, the exact location of these points is not crucial as long as they stay off the ROA. Since the Zubov equation provides a global condition (see loss function $L_{pde}(\theta)$ and $L_{data}(\theta))$, the data points will still provide information even though they are outside of ROA.
>
> ***$\omega$ in Theorem 2.4.***
> We indeed made the assumption $\omega(x) = ||x||^p$, but we mistakenly deleted it. We will make sure to add it back.
>
> ***Loss function.*** The loss function is a weighted combination of the loss terms as in Eq. (16) - (20) as follows:
>
> $L(\theta) = a_1 L_{zero}(\theta) + a_2 L_{pde}(\theta) + a_3 L_{data}(\theta) + a_4 L_{controller}(\theta) + a_5L_{boundary}(\theta)$
>
> where we have
>
> * $L_{zero}(\theta) = ReLU(V_\theta^2(0)-0.01)$. This term **minimizes $V_\theta(0)$ akin to Theorem 2.3**. We use a hinge loss to avoid potential numerical issues in the following CEGIS loop.
> * $L_{pde}(\theta) = \frac{1}{k_1} \sum (\nabla V_\theta(x_i)^\top f(x_i, detach(u(x_i))) + a(1+V_\theta(x_i))(1-V_\theta(x_i))||x_i||^p)^2$. This term **minimizes the PDE residual according to the Zubov equation in Theorem 2.4**. We detach the controller to let the Zubov Theorem only shape the Lyapunov function.
> * $L_{data}(\theta) = \frac{1}{k_2} \sum (V_\theta(x_i) - \tanh(a\int_0^T ||\phi(t,x)||^p dt) - \tanh^{-1}(V_\theta(\phi(T,x))))^2$. This term further **trains the Lyapunov function according to equation (10) in line 188**.
> * $L_{controller}(\theta) = \frac{1}{k_1}\sum detach(\nabla V_\theta (x_i))^\top f(x_i, u(x_i))$. This term **encourages the learned controller $u$ to follow a Sontag-style formula** with the current Lyapunov function.
> * $L_{boundary}(\theta) = \frac{1}{k_3} \sum (V_\theta(y_i) -1)^2$. Here the $y$'s are chosen from $\partial(2\Omega)$ where $\Omega$ is our training domain. This term **regularizes early-stage training** by driving $V_\theta(y_i)$ to $1$, and thus prevents the trajectory from diverging too early as $L_{controller}(\theta)$ trains the system to have decreasing Lyapunov function value along trajectories.
>
> We will move the loss function discussion to the main text and provide these intuitions to ensure the paper is self-contained.
>
>
> ***CEGIS Loop.*** In Algorithm 1 line 13, we construct a **counterexample dataset**, which is then used in line 15, where the loss function $L_{cegis}$ is evaluated on these counterexamples. We will revise the notation for the loss functions $L_{cegis}(\theta)$ as $L_{cegis}(\theta;x)$ and uses $L_{cegis}(\theta;\text{Dataset})$ in Algorithm 1 line 15.
>
> ### **W2 & Q1: Clarifications on α,β-CROWN**
>
> We would like to clarify that we never mentioned that we used Interval Bound Propagation (IBP) in our paper. Instead, we used tighter linear bound propagation. We did not provide a detailed discussion of α,β-CROWN in our paper as it is a well-developed tool, and not our main contribution. α,β-CROWN is a GPU-accelerated neural network verifier capable of handling complex computation graphs with general nonlinearities. Intuitively, CROWN can be seen as a generalization of backpropagation. Just like how gradients are propagated through a network, CROWN propagates **linear upper/lower bounds** across the entire computation graph over an input region. Instead of computing derivatives, it computes **linear relaxations** for each nonlinear operation. These local relaxations are then composed layer by layer (analogous to how backpropagation works), resulting in global affine bounds of output with respect to inputs. This approach, known as **linear bound propagation**, is **much tighter than IBP**, which only tracks independent input-output intervals and ignores inter-variable dependencies. Moreover, CROWN can do branch-and-bound to further tighten the bounds. By applying CROWN to our computation graph $\nabla V(x)f(x)$, we can provide formal verification in a more scalable manner compared to traditional SMT solvers like dReal.
>
> While the linear bound propagations have been presented in α,β-CROWN’s original paper [1] and extended to general computation graphs [2], we understand that it can be challenging to grasp these algorithms quickly, so we will **provide a toy example** on how it applies to our verification setting in the appendix.
>
> Although not our primary focus, our work does make new technical extensions to α,β-CROWN, including enabling support on computation graphs with Jacobian. Obtaining the **tightest possible** relaxations for nonlinearities with Jacobian operators is also challenging - the differentiation of many operators, such as tanh or sigmoid, can introduce some tricky nonlinearities like $1 - \tanh^2(x)$. In principle, α,β-CROWN could handle this by treating it as a combination of operators, like treating $\tanh^2(x)$ as the combination of $\tanh(x)$ and $x^2$, and relaxing it using the combination of linear bounds of these more elementary nonlinearities. However, this bound will be loose. In our work, we implement a **tight relaxation** of the function $h(x) = 1 - \tanh^2(x)$ (see Appendix B.2) directly to significantly tighten the bound of this operator (**over 6 times tighter**), and apply the same idea to sigmoid’s derivative. We discussed these in Appendix B.2.
>
> ### **W3: Guarantees near origin**
>
> The paper mentioned by the reviewer addresses only **polynomial systems and activations**, whereas our approach needs to handle **general nonlinearities**. In our setting, achieving exact guarantees is impossible due to verifiers’ numerical issues near the origin when handling general nonlinearities. To our knowledge, no prior work can achieve hole-free verification in this setting. Therefore, we instead provide alternative **theoretical guarantees** ensuring the excluded region is **forward-invariant**. In comparison, many previous works do not provide any guarantee at all, as discussed in lines 228-230. Moreover, the actual region excluded is **tiny** and thus our approach remains **practically effective**. For example, $\lbrace V(x) \le 0.0068\rbrace$ is removed for the Cartpole system, which occupies **only 0.003% of the ROA**. Appendix C.4 shows that our hole excluded is **negligible** for all tested systems.
>
> ### **W4 & Q2: High-dimensional Verification**
>
> We will revise our paper to be clear about which systems are verified under which schemes. Moreover, we will change our terminology used in section 3.2. Namely, we will change "Verification Schemes" into "Evaluation Schemes", and change "Empirical PGD Verification" and "Monte-Carlo Trajectory Verification" to "PGD Evaluation" and "Trajectory Evaluation" to avoid misunderstandings. We note that although we could only verify forward invariance for 6D systems, this is **already the SOTA result for continuous-time systems**. Moreover, we achieve **significantly larger ROA than all the baselines**, which inherently makes the verification problem harder. Below, we list the evaluation schemes we apply for each system and the ROA volume improvement compared to the previous SOTA.
>
> |Dimensions|Systems|Traj.|PGD.|Formal Verification|ROA (compare to SOTA)|
> |:-:|:-:|:-:|:-:|:-:|:-:|
> |2D|Van Der Pol|✓|✓|Full|2.85x|
> ||Double Int|✓|✓|Full|2.32x|
> ||Pendulum B|✓|✓|Full|6.05x|
> ||Pendulum S|✓|✓|Full|3.82x|
> ||Path Tracking B|✓|✓|Full|5.07x|
> ||Path Tracking S|✓|✓|Full|4.97x|
> |4D|Cartpole|✓|✓|Full|330.3x|
> |6D|2D Quadrotor|✓|✓|Forward Invariance ($\lbrace 0.2 \le V \le 0.21\rbrace$)|150000x|
> ||PVTOL|✓|✓|-|383x|
> ||Ducted Fan|✓|✓|-|N/A|
> |12D|3D Quadrotor|✓|-|-|N/A|
>
> The reviewer mentions that some SMT baselines, such as NLC and Fossil, could scale on par with our method. We respectfully disagree and believe our method is much better than SMT baselines. We list our reasons below.
>
> - NLC and Fossil [3] have **soundness issues**. They mistakenly always use a bias-free network for controllers, leaving $u(0) = 0$. However, this is not true for Path Tracking and all >6D systems we considered. Further discussions can be found in Appendix C.1.
> - NLC and Fossil **do not scale to Cartpole system** that we excel in. Moreover, we have compared our results to Fossil on the systems that do not have soundness issues, which have been shown to be much better than NLC [3]. Our method achieves greater scalability and much larger ROA volumes compared to Fossil.
>
> |Systems|Fossil 2.0||Ours||
> |:-:|:-:|:-:|:-:|:-:|
> ||ROA|Succ|ROA|Succ|
> |Van Der Pol|3.05±0.58|60%|**57.6±3.4**|100%|
> | Double Int|258.1±36.3|80%|**302.5±10.7**|100%|
> | Pendulum S|106.5 ± 23.5|80%|**1169.2±124.5**|100%|
> | Pendulum B|285.2±53.3|80%|**2946.5±149.1**|100%|
> | Cartpole|—|0%|**306.1±54.2**|100%|
>
> ### **References**
> [1] Zhang, Huan, et al. "Efficient neural network robustness certification with general activation functions." NeurIPS 31 (2018).
> [2] Xu, Kaidi, et al. "Automatic perturbation analysis for scalable certified robustness and beyond." NeurIPS 33 (2020).
> [3] Edwards, Alec, et al. "Fossil 2.0: Formal certificate synthesis for the verification and control of dynamical models."​​HSCC. 2024.

---

> > ### Comment · Reviewer_Pjvg · 2025-08-03
> >
> > Thank you for the detailed response! The response addresses and clarifies all my questions and I do feel more positive about the paper now, especially with the question about which experiments have been formally verified being clarified. I have increased my score accordingly. I strongly recommend to include these clarifications in the final version of the paper.

---

> > > ### Author Response · Authors · 2025-08-04
> > >
> > > Thank you for your response! It is great to know that your concerns are well clarified and you feel more positive about our work. We will make sure to reflect all the clarifications made here in our revised manuscript to ensure there is no miscommunication. We are happy to take in any further questions you have regarding our work!

---

### Decision · Program_Chairs · 2025-09-17

**Decision:**

Accept (spotlight)

**Comment:**

The paper proposes a method that jointly learns a neural control policy and a neural Lyapunov function, using a novel Zubov-inspired loss to explicitly maximize the region of attraction (ROA). Experiments show that this approach achieves larger ROAs than prior methods while ensuring formal correctness. Authors present several practical enhancements to alpha-beta CROWN, including linear relaxations of Jacobian operators and a branch-and-bound heuristic, improving the scalability and efficiency of verification. The experimental benchmarks span a wide range of complexity and dimensionality, clearly demonstrating the benefits of the proposed approach relative to existing baselines. One weakness at the moment is scaling to higher dimensional systems. Overall, the paper is a strong contribution. It is strongly suggested that the authors consider the suggestions for improving the presented by the reviewers.